# Collaborative Gold Mining Algorithm: An Optimization Algorithm Based on the Natural Gold Mining Process

Alireza Salehan [1,*] and Bahman Javadi [2]

1 Department of Computer Engineering, University of Torbat Heydarieh, Torbat Heydarieh 9516168595, Iran
2 School of Computer, Data and Mathematical Sciences, Western Sydney University, Penrith, NSW 2751, Australia
* Correspondence: salehan@torbath.ac.ir

**Abstract:** In optimization algorithms, there are some challenges, including lack of optimal solution, slow convergence, lack of scalability, partial search space, and high computational demand. Inspired by the process of gold exploration and exploitation, we propose a new meta-heuristic and stochastic optimization algorithm called collaborative gold mining (CGM). The proposed algorithm has several iterations; in each of these, the center of mass of points with the highest amount of gold is calculated for each miner (agent), with this process continuing until the point with the highest amount of gold or when the optimal solution is found. In an n-dimensional geographic space, the CGM algorithm can locate the best position with the highest amount of gold in the entire search space by collaborating with several gold miners. The proposed CGM algorithm was applied to solve several continuous mathematical functions and several practical problems, namely, the optimal placement of resources, the traveling salesman problem, and bag-of-tasks scheduling. In order to evaluate its efficiency, the CGM results were compared with the outputs of some famous optimization algorithms, such as the genetic algorithm, simulated annealing, particle swarm optimization, and invasive weed optimization. In addition to determining the optimal solutions for all the evaluated problems, the experimental results show that the CGM mechanism has an acceptable performance in terms of optimal solution, convergence, scalability, search space, and computational demand for solving continuous and discrete problems.

**Keywords:** collaborative gold mining; optimization algorithms; continuous mathematical functions; optimal placement of resources; traveling salesman problem (TSP); bag-of-tasks scheduling (BoT)

## 1. Introduction

Several optimization algorithms have been proposed to find (near-) optimal solutions to complex problems in different areas of science and technology. As many of these algorithms are inspired by processes in nature or biological behaviors, they simplify the complex operation of problems and provide viable solutions. The success of such algorithms has led to the introduction of a lot of nature-inspired and bio-inspired optimization methods [1,2].

There are different types of problems that can be solved with optimization algorithms: discrete, continuous, combinatorial, and multi-objective problems [3,4]. In all of the proposed optimization methods, the behavior of a procedure in nature and biology, or even human social behavior, is examined and mapped to the algorithm steps. Due to their excellent efficiency, these optimization methods may be applied to many real-world problems in different contexts. They are able to solve problems dealing with computer networks [5], power systems [6], telecommunications [7], intrusion detection [8], data mining [9], face recognition [10], clustering [11], transportation [12], and robotics [13].

While there are many optimization algorithms, there are still some challenges that keep researchers motivated to introduce new optimization algorithms. Some of these challenges includes lack of optimal solution (some methods cannot find an optimal or suboptimal

solution to a problem), slow convergence (in some methods, obtaining the optimal solution requires a large number of iterations), lack of scalability (some methods cannot work properly for large dimensional problems), partial search space (some methods do not consider the entire search space to find the global optimal solution; instead, they are only able to find the local optimal solution), and high computational demand required (some methods require a lot of computing resources to determine the optimal solution). These challenges make the optimization algorithm not applicable in real-world environments and on low-power devices.

With a focus on gold exploration and exploitation processes, this paper introduces collaborative gold mining (CGM), a new optimization algorithm. With the participation of a number of gold miners (optimization problem agents), the proposed algorithm can discover the position with the highest amount of gold (optimal solution) in an entire search area of an n-dimensional geographical space. The CGM algorithm includes several iterations, and in each iteration, the center of mass of points with the highest amount of gold is calculated for the miners (exploration process); each of these repeats this process based on the new position so as to extract more gold (exploitation process). This process continues until the location with the highest amount of gold, the optimal solution, is determined. The CGM mechanism can be applied to both minimizing and maximizing problems.

After the theoretical expansion of the proposed mechanism, we show its applicability and efficiency for discrete and continuous problems. This is achieved with a set of benchmark functions. Then, the outputs are compared with other famous optimization mechanisms. These comparisons indicate that the proposed mechanism is efficient and scalable and that it can even outperform other existing algorithms for the sample problems.

Some important significances of the proposed CGM mechanism are the following:

- In addition to determining the optimal solution, the CGM mechanism can provide better efficiency in terms of convergence, scalability, and search space.
- The CGM mechanism can determine the appropriate solution with the least number of members of the population.
- For most problems, this method converges to a quasi-optimal or optimal solution in the same initial iterations.
- The CGM mechanism is also able to solve problems on different scales.
- This method does not limit the search space during iterations and tries to find the optimal solution in the whole search space during all iterations.

The rest of the paper is organized as follows: Section 2 presents the review of a related survey in the field of optimization algorithms inspired by human social behavior and lifestyle. Section 3 reports the investigation and mathematical formulation of the mechanism of collaborative gold mining based on the concept of the center of mass. This section also presents details on the CGM mechanism and its algorithm pseudocode. Section 4 presents the evaluation of the performance of the proposed optimization algorithm with the implementing of several discrete and continuous practical problems. These problems include a series of continuous mathematical functions as well as practical problems addressing the optimal placement of resources, the traveling salesman problem, and bag-of-tasks scheduling. The evaluated criteria include optimal solution, convergence, scalability, search space, and computational demand. Finally, Sections 5 and 6 present discussions and conclusions, respectively.

## 2. Related Work

Currently, several optimization algorithms have been proposed that have drawn upon different sources of inspiration [1]. Despite their diversity in properties and characteristics, the basis of most optimization algorithms inspired by nature or biology is found in one of five classic and famous algorithms, namely, ant colony optimization [14], artificial bee colony [15], particle swarm optimization [16], genetic algorithm [17], and differential evolution algorithm [18]. Most proposed algorithms assume some of the properties and characteristics of these five basic algorithms. Due to the large number of optimization

algorithms, it is beyond the scope of this paper to discuss all of them; more information can be found in other studies [1,2,19–21].

In addition to nature-inspired and bio-inspired mechanisms, there are some optimization methods that operate based on human social behavior and lifestyle. The most important and well-known of these are the imperialist competitive algorithm [22], the brain storm optimization algorithm [23], the anarchic society optimization algorithm [24], the ideology algorithm [25], the pop music algorithm [26], soccer game optimization [27], the golden ball algorithm [28], and FIFA world cup competitions [29].

In the imperialist competitive algorithm [22], countries, as a population of the optimization problem, are divided into colonies and imperialists and are initially located within a number of empires. Then, among these empires, an imperialist power struggle begins. The power of each empire is calculated based on the power of the imperialist plus a percentage of the average of its colonies' power. In addition, countries are able to switch and join other empires. During this competition, weak empires fall and powerful empires overcome the colonies of weak empires. This competition continues until just one empire and its colonies stay as the strongest entity. The last standing empire is the solution to the optimization problem. In contrast, the brain storm optimization algorithm [23] is inspired by the process of human brainstorming. This inspiration derives from the fact that some problems in the real world cannot be overcome by one person and humans must work together to solve them. In this algorithm, individuals are considered as the population of the optimization problem and based on convergent and divergent operations, they present the perspective of the problem and are grouped in the search space in order to produce the best solution during iterations.

The anarchic society optimization method [24] is inspired by the social grouping of human beings. In this method, those having fickle, unstable, adventurous, and irrational behavior are considered as members of one social group. To improve their situation, these people should exhibit anarchic behavior. As the level of difference among the members' situations grows, the level of anarchic behavior among individuals intensifies. These anarchic behaviors are repeated until the whole problem space is explored; finally, an optimal solution is found. The ideology algorithm [25], in comparison, is based on the utilitarian and competitive behavior of individuals who are members of different political parties. By following the ideology of their local leader and forming a close relationship with the leader, political party members can improve their ranking. Since the behavior of other political party leaders can be examined and compared with that of their own, there is an incentive to change parties. Furthermore, each of the local party leaders compare themselves to each other in vying to become the global leader. In this competitive ranking-based algorithm, each individual is identified as a problem solution and the global leader is defined as the optimal problem solution.

The pop music algorithm [26] can solve various combinatorial optimization problems by analyzing concepts used in heuristic methods. The main idea of this method is to locally optimize the subdivisions of a solution until the optimal solution of the problem is reached. The soccer game optimization mechanism [27] is a population-based collaborative optimization method inspired by the concepts of football. This mechanism is based on football players' actions on the field: move off, as the exploration process, and move forward, as the exploitation process. In this mechanism, the position of each player is considered as one of the initial solutions to the optimization problem, and the performance of each player, as an objective function, is evaluated based on this position. Among all the members of the team, the player with the best position is known as the owner of the ball or the dribbler. The dribbler tries to move forward, pass the ball to other players, all the while being followed by the other players. The dribbler's "move forward" action continues until he reaches the point of his own maximum efficiency (optimal solution) and, as a result, also that of the whole team's (in other words, the opponent's goal).

The golden ball algorithm [28] first considers a population of players and coaches and then divides them into different teams. Each team has power based on the performance of

its members, with this power acting as the objective function of the optimization problem. Each season, these teams compete independently, holding competitions and performing training exercises. At the end of each season, the transfer phase begins, during which players and coaches switch among teams. The process of repeating seasons and competing continues until team members become reluctant to transfer. At this time, the team with the most power is the ultimate solution to the optimization problem. Finally, [29] introduced an optimization technique that finds the optimal solution based on the FIFA World Cup games. In this technique, each of the teams represents a country, as the initial solution to the problem, and competes with its neighbors. The top two teams on each continent progress to the World Cup. The selected teams then compete in the final games to determine which is the best team in the world as the optimal solution to the optimization problem.

While there are a lot of optimization mechanisms, there are still a number of challenges including lack of optimal solution, slow convergence, lack of scalability, partial search space, and high computational demand. The existence of these challenges leads to the impossibility of using optimization algorithms in some real applications on low-power devices and equipment. One of these applications includes recommending useful services or devices in a smart environment for an IoT platform [30,31]. In the present paper, we propose a new optimization method to address some of these challenges based on collaborative gold mining.

## 3. Materials and Methods

We consider a number of gold miners who collaboratively search for gold in a specific geographic area. The search operation consists of exploration and exploitation stages, with the miners aiming to find the location with the highest amount of gold in the whole geographical area. Collaborative mining means that a miner, after finding a new location with gold, provides that position's information to other miners if it holds more gold than previously discovered positions. In this way, miners can follow a better path of discovery that leads to a location with the highest amount of gold. In this mechanism, the geographic search area is assumed to be an n-dimensional area, in which each miner is able to alter any of the dimensions when moving from the current position to a new position.

In order to cover the whole area, the miners are randomly scattered throughout the geographical area, where the mining operation is not limited to a specific part of the geographic space. Once the miners are dispersed, the collaborative mining operation begins. In the first search, the miners share information about the amount of gold found, which identifies the positions where more gold was discovered. Out of all the results obtained from the first search, the top three positions reporting the highest amount of gold are called the first-best, second-best, and third-best positions. As their name suggests, these three positions represent the coordinates of the first, second, and third points, respectively, which have the highest amounts of gold among all the other positions.

Based on the miners' current position and the values of the three best positions, the center of mass is calculated for each miner, where these new values indicate which positions should be explored and exploited next. Since it may not be possible to explore the given positions (for example, due to the inability to dig or inaccessibility of the new position), the miner is able to explore near or around that position instead of that exact point. Figure 1 illustrates how to determine the next position based on the miner's current position values and the three best positions.

After the identification of new mining positions using the center of mass, the second stage is to explore and exploit gold at these positions and to determine the amount of gold discovered there. Then, based on the results obtained from this stage, the values of the first-best, second-best, and third-best positions are updated to determine the next positions to mine. This process will continue as long as the miners are willing to collaborate; finally, the first-best position, indicating the location with the most gold, will be revealed.

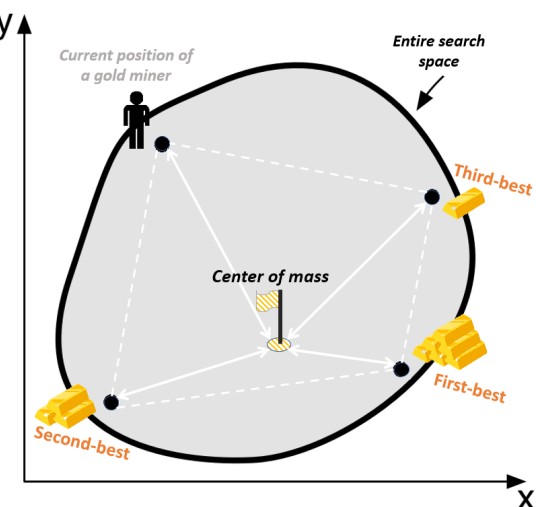

**Figure 1.** Determination of the next position based on the values of a miner's current position and the three best positions.

### 3.1. Center of Mass

In mechanics, the center of mass or balance point is a unique point, at which the sum of the weighted relative position vectors of distributed masses is zero. In other words, if several particles with specific masses are distributed in space, the center of mass of these particles indicates their balance point. At the center-of-mass point, the sum of the torques in a clockwise direction equals the sum of the torques in a counterclockwise direction around this point [32].

Let us consider $n$ particles, $P_i$ ($i = 1, 2, \ldots, n$), in a three-dimensional space whose mass is $m_i$ and that the position vector of each is $\vec{r_i}$. The center of mass of these $n$ particles, which is obtained from the following equation, is point $\vec{r_{CM}}$.

$$\vec{r_{CM}} = \frac{m_1\vec{r_1} + m_2\vec{r_2} + \cdots + m_n\vec{r_n}}{m_1 + m_2 + \cdots + m_n} = \frac{\sum_{i=1}^{n} m_i\vec{r_i}}{M} \tag{1}$$

In Equation (1), $M$ is the final mass of all particles. In this three-dimensional space, because the position vector of each particle has three values ($x$, $y$, and $z$), the center of mass of each dimension can be calculated separately. For this purpose, Equation (2) is used as follows:

$$x_{CM} = \frac{1}{M} \sum_{i=1}^{n} m_i x_i$$
$$y_{CM} = \frac{1}{M} \sum_{i=1}^{n} m_i y_i \tag{2}$$
$$z_{CM} = \frac{1}{M} \sum_{i=1}^{n} m_i z_i$$

In other words, to calculate the center of mass of several points in a space with the desired number of dimensions (even more than three dimensions), Equation (2) is sufficient. Figure 2 illustrates the center of mass (CM) for four hypothetical points with different masses in a two-dimensional space. In this figure, Equation (2) calculates the center-of-mass value for each dimension. In addition to the applications of the center of mass in physical and mechanical sciences, this concept offers many applications in other fields, such as the manufacturing of medical equipment, the design of sports equipment, and improvements in daily life [32,33].

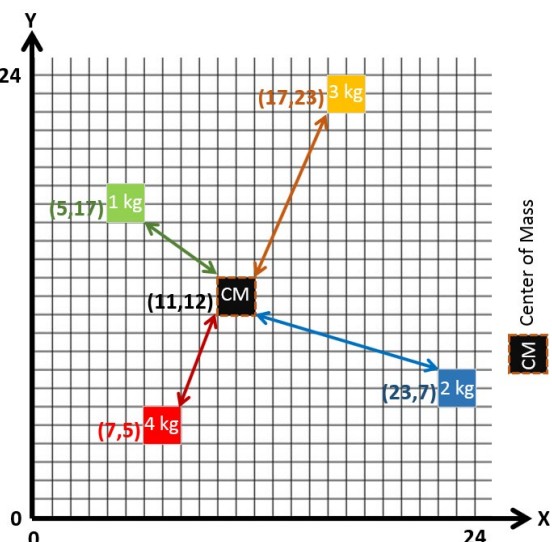

**Figure 2.** Center of mass of four points with different masses in a two-dimensional (2D) space.

### 3.2. Collaborative Gold Mining Optimization Mechanism

The collaborative gold mining (CGM) optimization mechanism is able to solve both maximization and minimization problems. The goal in maximization problems can be finding the position with the highest amount of extracted gold; in minimization problems, the goal may be locating the position with the lowest level of impurity in the extracted gold. In the CGM mechanism, a population of gold miners is initialized as optimization agents at random positions across the intended geographic area. This search area is assumed to be an n-dimensional area; therefore, the number of optimization problem variables are *n*, each of which can be set in a specific range. After these variables are set for each gold miner as the problem population, the optimization objective function is calculated for each one in order to determine the values of the three best positions. In the CGM mechanism, the number of population members is constant in all iterations.

Depending on the results of the objective function, the first three best values are selected as the values of the first-best, second-best, and third-best positions. According to the type of problem (maximization or minimization), the values of these three positions indicate the three points with the highest amount of gold or the lowest level of impurity, respectively. The values of these three positions are updated during the iterations of the algorithm; finally, after all the iterations are performed, the first-best position with the most gold (In maximization problems) or the lowest level of impurity (in minimization problems) in the whole search space is considered as the solution to the optimization problem.

After initializing the population and determining the values of the three best positions, the iterations of the algorithm are performed. The algorithm can include *m* iterations, in which the new position of each gold miner is first determined (exploration process). To obtain this new position, the center of mass is calculated for the miner's current position and the three best positions, with the result summed with a random value. This random value is considered because the miner may not mine the exact point determined by the center of mass (for reasons such as the inability to dig or inaccessibility of the new location). The random value is calculated differently in continuous and discrete problems, as discussed in the next section. After determining the new positions for all miners, the amount of gold obtained from the new points is calculated (exploitation process). If necessary, the values of the three best positions are updated so that the location of the points with the highest amount of gold or the lowest level of impurity is maintained in the three best variables.

It should be noted that, in each iteration of the algorithm and for each miner, the *n* variables of the problem are updated to finally determine the new position of each miner, which is achieved by calculating the objective function for these *n* updated variables. Equation (3) demonstrates the calculation of a gold miner's new position based on the

center-of-mass concept (Equations (1) and (2)) and using the values of the current position and the three best positions.

$$newPos_j^i = \frac{currentPos_j^i \times f_i^* + \alpha_j \times f_\alpha^* + \beta_j \times f_\beta^* + \gamma_j \times f_\gamma^*}{f_i^* + f_\alpha^* + f_\beta^* + f_\gamma^*} + \varphi \tag{3}$$

In Equation (3), the value of *i* represents the *i*-th miner and the value of *j* indicates the variable *j* of the *i*-th miner, so that $1 \leq j \leq n$. Moreover, `newPos` and `currentPos` indicate the new and current positions of the variable *j* of the *i*-th miner, respectively. $f^*$ is the optimization objective function, and $\alpha$, $\beta$, and $\gamma$ are the values of the first-best, second-best, and third-best positions, respectively. Index *j* in each of the values of $\alpha$, $\beta$, and $\gamma$ represents variable *j* of these three positions. In addition, $\varphi$ is a random number, as explained in Section 3.3. Note that, for $\theta \in \{i, \alpha, \beta, \gamma\}$, function $f_\theta^*$ calculates the optimization objective for the miner or for one of the three best positions based on the values of all variables.

The other iterations are performed similarly to the first iteration, where each miner's new position is first found and then, if necessary, the values of the three variables, first-best, second-best, and third-best, are updated. As mentioned above, after all iterations are completed, the first-best position is the optimal solution for the optimization problem. This final first-best position shows the location of the point with the highest amount of gold or the lowest level of impurity among all the points mined (according to Equation (3), at the end of the iterations, $\alpha$ represents the values of the variables in the optimal position, and $f_\alpha^*$ is the solution to the optimization problem).

### 3.3. Determination of Random Value $\varphi$

Equation (3) calculates the new position of each gold miner according to the current position of the miner and the three best positions. Equation (3) is the sum of the center of mass and random number $\varphi$. This random number is considered because the miner may not be able to mine the exact point obtained from the center-of-mass equation. Determining random number $\varphi$ is performed differently in continuous and discrete problems.

In continuous problems, the variables can be defined in the specified range and can change continuously to any desired extent. However, in the most discrete problems, variables can only be a permutation of the permissive integer values and so only these acceptable values can be switched among the variables. In other words, unlike continuous problems, the value of a variable in most discrete problems cannot be changed arbitrarily, and only different permutations of the permissive integer values can be assigned to the problem variables.

If the problem is continuous, the permissive range for moving the miner in each dimension of the space must be chosen between the minimum and maximum values of the three best positions in that dimension. Suppose that the miner intends to find his movement value in the *j*-th dimension. Assuming that $CM_j$ is the center of mass and that values $\alpha_j$, $\beta_j$, and $\gamma_j$ are the values of the three best points in the *j*-th dimension, the range for moving the miner is $\left(\min(\alpha_j, \beta_j, \gamma_j) - CM_j, \max(\alpha_j, \beta_j, \gamma_j) - CM_j\right)$, and a random number with a normal distribution in this range can be considered as the acceptable threshold value, $\varphi$ (miner movement value). The reason for selecting this range is because the center of mass is among the best points; therefore, mining for gold outside the intersection of these three points cannot provide a better result for the miner, at least in the current iteration. The two points, $\min(\alpha_j, \beta_j, \gamma_j) - CM_j$ and $\max(\alpha_j, \beta_j, \gamma_j) - CM_j$, are not considered in the above range because the new position must be different from the three best points in the *j*-th dimension.

If the problem is discrete, however, $\varphi$ can be a random number selected from the selectable and allowable integer values. Let us suppose that the problem has *n* variables, so that the values of $v_i$ ($i = 1, 2, \ldots, n$) can be assigned to each of these variables. Considering the permissive values of $v_i$, $\varphi$ can be a random number from these *n* values. Since it may not be possible to assign duplicate values to variables in some problems (such as task

scheduling and traveling salesman problems), the initialization of variables in these types of problems is a permutation of $v_i$ values. In this case, random number $\varphi$ can be selected from any of these $n$ values, provided that the new position obtained from Equation (3) is not a duplicate. In the event of duplication, the procedure for resetting $\varphi$ from the values of $v_i$ continues until the result of Equation (3) generates a new position that has not been previously selected by the gold miner for the intended variable.

### 3.4. The Proposed CGM Algorithm

Algorithm 1 describes the CGM optimization for both maximization and minimization problems, including continuous and discrete types. Table 1 presents some important variables and parameters utilized in the CGM mechanism. This table shows the concept inspired by the collaborative gold mining process and the description of the parameters in the optimization problem space.

**Table 1.** Important variables and parameters used in the CGM algorithm.

| Name | Concept Inspired by the Collaborative Gold Mining Process | Description in the Proposed CGM Algorithm |
|---|---|---|
| Iters | Number of recurrences of the mining process | Number of iterations |
| nPop | Number of collaborative gold miners | Number of populations (solutions) in all iterations |
| pop | Gold miners in all recurrences | The set of solutions in all iterations |
| nVar | Number of positions | Variables of the optimization problem |
| LB | Lower bound of positions | Minimum value of each variable |
| UB | Upper bound of positions | Maximum value of each variable |
| $\alpha$ | Location of first-best position with the highest amount of gold (or the lowest level of impurity) extracted | The first-best solution in all iterations |
| $\beta$ | Location of second-best position with the highest amount of gold (or the lowest level of impurity) extracted after the first-best location | The second-best solution in all iterations |
| $\gamma$ | Location of third-best position with the highest amount of gold (or the lowest level of impurity) extracted after the second-best location | The third-best solution in all iterations |
| CGM | Amount of extracted gold (or extracted impurities), shown using $f^*$ in Equation (3) | Objective function |
| $\varphi$ | Random amount of movement for each miner | Acceptable threshold for determination of the new position |

The inputs of this algorithm are the values of nPop, nVar, Iters, LB, and UB, and the only objective output of the algorithm is the optimal solution, which is returned by variable $\alpha$. This variable includes the location and objective function of the point with the highest amount of gold extracted (for maximization problems) or the lowest level of impurity extracted (for minimization problems) as the solution to the optimization problem during all the iterations. Lines 1 to 22 initialize the members of the problem population and the values of the three best positions. In the first three lines of the algorithm, the initial values of the three variables, $\alpha$, $\beta$, and $\gamma$, as the three best positions, are considered NULL. Since each member of the population, along with the three best positions, contains two fields, Pos (position vector) and Obj (value of the objective function), placing NULL in the three best positions means that at the beginning of the algorithm, the value of the Pos field is 0 and the value of the Obj field is assumed to be $-\infty$ (for maximization problems) or $+\infty$ (for minimization problems).

If the problem is continuous, for each member of the population in lines 5 to 7, nVar random numbers are generated in the range of [LB, UB] and stored as the initial locations of the gold miners in the pop array. However, if the problem is discrete, a random permutation of the problem variables is recorded in lines 8 to 10 as the initial location for each member of the population. In line 11, the value of the objective function is calculated for each member

of the pop population, and this value is compared with the values of the three best positions in lines 12 to 21 to update the value of these three positions, if necessary. Since Algorithm 1 is written for both maximization and minimization problems, operator $\gtrless$ can include two values, > and <, for maximization and minimization problems, respectively.

In lines 23 to 55, the iterations of the CGM algorithm are performed, where lines 26 to 42 describe the exploration process and lines 43 to 53 explain the exploitation process of the CGM mechanism. In each iteration and for each member of the population, all problem variables are first updated. In line 26, the center-of-mass value for variable $j$ of the $i$-th member of the population is calculated, and the result is placed in CM. If the problem is continuous, the two variables, A and B, are set in lines 28 and 29 based on the value of CM, as well as the minimum and maximum values for each of the three best positions for the given variable, and in line 30, a random number with the normal distribution in the range between A and B is calculated as the value, $\varphi$. However, if the problem is discrete, in line 33, a random integer in the range of [0,nVar] is calculated and assigned to $\varphi$.

In line 35 of the algorithm, the new value of the variable is calculated based on Equation (3); then, lines 36 to 41 examine if the value is within the range. Since the new value for discrete problems must be in the range of nVar values, this value is first converted to an integer value in line 40 and then mapped to the range using the modulo operator (mod). After all the variables of each member of the pop population are updated, the value of the objective function is calculated in line 43. In this line, the CGM function computes the objective value of the problem. This value is then compared with the values of the three best positions in lines 44 to 53 so that the value of these three positions is replaced with the value of the objective function obtained for the population member, if necessary. Finally, the output of the algorithm is the first-best position that is returned as the optimization problem's optimal solution.

---

**Algorithm 1** The proposed **CGM** optimization algorithm

---

**Inputs:** The values of nPop, nVar, Iters, LB, and UB

---

      *// Initialization of population and three best locations*

1     $\alpha$ = NULL;

2     $\beta$ = NULL;

3     $\gamma$ = NULL;

      *// Population*, $\alpha$, $\beta$, and $\gamma$ *include two fields: Pos (positions) //*

      *and Obj (objective), in which:*

      *// \*.Pos = 0 and \*.Obj = $-\infty$ (for maximization problems) or*

      *// $+\infty$ (for minimization problems);*

4     for (i = 1; i $\leq$ nPop; i ++)

5       if (Problem type = = Continues)

6         $pop_i.Pos$ = LB $\leq$ nVar numbers of random values $\leq$ UB;

7        endif

8       if (Problem type = = Discrete)

9        $pop_i.Pos$ = Random_Permutation (nVar);

10      endif

11    $pop_i.Obj$ = CGM ($pop_i.Pos$);

12    if ($pop_i.Obj \gtrless \alpha.Obj$)

13      $\gamma \leftarrow \beta$ ;

14      $\beta \leftarrow \alpha$ ;

15      $\alpha \leftarrow pop_i$ ;

16    elseif ($pop_i.Obj \gtrless \beta.Obj$)

17            $\gamma \leftarrow \beta$ ;

18            $\beta \leftarrow pop_i$ ;

19         elseif ($pop_i.Obj \gtrless \gamma.Obj$)

20            $\gamma \leftarrow pop_i$ ;

**Algorithm 1** *Cont.*

```
21        endif
22      endfor_i
        // Iterations of collaborative gold mining process
23      for (iter = 1; iter ≤ Iters; iter ++)
24        for (i = 1; i ≤ nPop; i ++)
25          for (j = 1; j ≤ nVar; j ++)
            // Exploration process
```
26          $CM = \dfrac{pop_i.Pos_j \times pop_i.Obj + \alpha.Pos_j \times \alpha.Obj + \beta.Pos_j \times \beta.Obj + \gamma.Pos_j \times \gamma.Obj}{pop_i.Obj + \alpha.Obj + \beta.Obj + \gamma.Obj}$ ;
```
27            if (Problem type = = Continues)
28              A = MIN (α.Posⱼ , β.Posⱼ , γ.Posⱼ) - CM;
29              B = MAX (α.Posⱼ , β.Posⱼ , γ.Posⱼ) - CM ;
30              φ = Normal_Random(A,B);
31            endif
32            if (Problem type = = Discrete)
33              φ = Integer_Random(0,nVar);
34            endif
35              newPos = CM + φ ; // Equation (3)
36            if (Problem type = = Continues)
37              popᵢ.Posⱼ = LB ≤ newPos ≤ UB ; // Checking inside of range
38            endif
39            if (Problem type = = Discrete)
40              popᵢ.Posⱼ = ⌈ newPos ⌉ mod nVar; // Checking inside of range
41            endif
42          endfor_j
          // Exploitation process
43        popᵢ.Obj = CGM (popᵢ.Pos);
44        if (popᵢ.Obj ≳ α.Obj)
45          γ ← β ;
46          β ← α ;
47          α ← popᵢ ;
48        elseif (popᵢ.Obj ≳ β.Obj)
49            γ ← β ;
50            β ← popᵢ ;
51          elseif (popᵢ.Obj ≳ γ.Obj)
52                γ ← popᵢ ;
53        endif
54      endfor_i
55    endfor_iter
```

**Objective output:** The optimal solution $\alpha$

*Note*: Operator ≳ is replaced by > for *maximization problems* and < for *minimization problems*.

## 4. Results

To evaluate the proposed CGM algorithm, we employed several continuous mathematical functions and several discrete and continuous practical examples. These functions and examples are NP-hard, and finding their optimal solutions is very costly. Some famous optimization algorithms, including the genetic algorithm (GA) [34], simulated annealing (SA) [35,36], particle swarm optimization (PSO) [37], and invasive weed optimization (IWO) [38], were used as the benchmark for performance evaluation. All codes were implemented with the C# programming language in the Visual Studio.NET 2019 framework. Moreover, all scenarios were performed on a machine with Windows 10 Pro that has an Intel 2.30 GHz Core i5-2410M CPU, 6.00 GB of RAM, and a 1 TB HDD. In Section 4.1, continuous mathematical functions and continuous/discrete application examples are introduced, and in Sections 4.2–4.5, the metrics for optimal solution value, convergence, scalability, search space, and computational demand are used to evaluate all the optimization techniques.

### 4.1. The Definition of Applied Continuous and Discrete Problems

4.1.1. Continuous Mathematical Functions

There are different mathematical functions utilized to evaluate optimization mechanisms. Table 2 lists nine mathematical functions selected for evaluating the CGM mechanism [39]. These continuous functions are the most common functions used for the evaluation of the optimization mechanisms. Moreover, due to the specific input range and the best solution value, these continuous functions make it easy to evaluate the proposed optimization algorithm. In addition to the function name, Table 2 presents other characteristics, including the number of dimensions, the mathematical definition, input range, best solution value, and the three-dimensional (3D) view of the function graph.

**Table 2.** List of selected continuous mathematical functions for evaluating the CGM algorithm.

| Function Name | Number of Dimensions | Mathematical Definition | Input Range | Best Solution Value | 3D View of Function Graph |
|---|---|---|---|---|---|
| Sphere | $n$ | $f(x) = \sum\limits_{i=1}^{n} x_i^2$ | $[-5.12, 5.12]$ | 0.0 |  |
| Griewank | $n$ | $f(x) = \sum\limits_{i=1}^{n} \frac{x_i^2}{4000} - \prod\limits_{i=1}^{n} \cos\left(\frac{x_i}{\sqrt{i}}\right) + 1$ | $[-600, 600]$ | 0.0 |  |
| ZeroSum | $n$ | $f(x) = \begin{cases} 0 & \text{if } \sum_{i=1}^{n} x_i = 0 \\ 1 + \sqrt{10000|\sum_{i=1}^{n} x_i|} & \text{otherwise} \end{cases}$ | $[-10, 10]$ | 0.0 |  |
| Rastrigin | $n$ | $f(x) = 10n + \sum\limits_{i=1}^{n} \left[x_i^2 - 10\cos(2\pi x_i)\right]$ | $[-5.12, 5.12]$ | 0.0 |  |
| Qing | $n$ | $f(x) = \sum\limits_{i=1}^{n} \left(x_i^2 - i\right)^2$ | $[-500, 500]$ | 0.0 |  |
| Zacharov | $n$ | $f(x) = \sum\limits_{i=1}^{n} x_i^2 + \left(\frac{1}{2}\sum\limits_{i=1}^{n} ix_i\right)^2 + \left(\frac{1}{2}\sum\limits_{i=1}^{n} ix_i\right)^4$ | $[-5, 10]$ | 0.0 |  |
| Plateau | $n$ | $f(x) = 30 + \sum\limits_{i=1}^{n} |x_i|$ | $[-5.12, 5.12]$ | 0.0 |  |
| Easom | 2 | $f(x) = -\cos(x_1)\cos(x_2)e^{-(x_1-\pi)^2-(x_2-\pi)^2}$ | $[-100, 100]$ | $-1.0$ |  |
| Matyas | 2 | $f(x) = 0.26\left(x_1^2 + x_2^2\right) - 0.48x_1x_2$ | $[-10, 10]$ | 0.0 |  |

### 4.1.2. Optimal Placement of Resources

The optimal placement of resources is a continuous problem with several real applications [40]. With the assumption that several devices are deployed in the surrounding environment, this problem aims to find the best location for connecting these devices so that the distance of the resource to all devices is the shortest. A shorter distance can decrease the cost of connecting the resource to all devices. In addition, it can increase the availability of the resource. In addition to the criterion of distance, there may be other important criteria depending on the type of application.

In this problem based on some geographical limitations, the unique shortest path cannot be used to calculate the distance, and the Manhattan distance is utilized [41]. The resource and devices are able to be located in an n-dimensional geographical space, which is also a continuous problem due to the continuity of geographical coordinates. When the number of devices or the size of the geographical space are increased, it is difficult and time-consuming to find a suitable location for a resource. As a result, this problem is an NP-hard problem. Equation (4) provides the objective function for this problem [39].

$$
\begin{aligned}
& min \text{ Distance} = \sum_{j=1}^{m} \sum_{i=1}^{n} \left| R_i - D_{ji} \right| \\
& \text{s.t.:} \\
& \forall k, j = 1, 2, \ldots, m \ (k \neq j) : \sum_{i=1}^{n} \left| D_{ki} - D_{ji} \right| \neq 0
\end{aligned}
\tag{4}
$$

In an n-dimensional space, let us assume that there is a resource that should be connected to an m number of devices. The location of this resource should be determined as a place, in which the resource is at the shortest distance from all devices. In this equation, i represents each of the dimensions in the coordinate axes ($i \in \{1, 2, \ldots, n\}$), and j is the device number ($j \in \{1, 2, \ldots, m\}$). The variables, $R_i$ and $D_{ji}$, are the coordinates of the i-th dimension of the resource and the i-th dimension of device j, respectively. Moreover, $\left| R_i - D_{ji} \right|$ refers to the absolute distance of the i-th dimension of the resource and the i-th dimension of device j, relative to each other. The only limitation of Equation (4) specifies that independent devices are deployed in separate locations, and this independence of deployment may be applied in only one dimension [39]. To solve this problem using the CGM mechanism, each resource placement was considered as a point featuring gold whose location was the geographical coordinates of the resource. The goal was to find the location with the minimum level of impurity in the extracted gold.

### 4.1.3. Traveling Salesman Problem

The traveling salesman problem (TSP) is a discrete problem that has been solved by many optimization mechanisms and has different applications in various industries [42]. In this problem, a traveling salesman starts his marketing campaign from one city and returns to the origin after meeting his customers in n other cities. In this problem, each city is visited only once, but the travel of the salesman from one city to another incurs an expense. The goal is to find a route in which all cities are traversed only once, thus producing the lowest cost for the salesman. The number of possible solutions for this problem is a permutation of n, because it is assumed that there is a unique path between each two cities. The aim is to find the route with the lowest travel cost between these *n*! possible modes. When the number of cities rises, the number of possible solutions increases. Based on this increase in the problem space, it can be said that TSP is NP-hard. Equation (5) indicates the objective function for this problem [39].

$$min \text{ Traveling Cost} = \sum_{i=1}^{n} \sum_{j=1}^{n} C_{ij} \times X_{ij}$$

s.t.:

$$X_{ij} = \begin{cases} 1 & if \ salesman \ moves \ from \ City_i \ to \ City_j \\ 0 & Otherwise \end{cases} \tag{5}$$

$$\sum_{i=1}^{n} X_{ij} = 1$$

$$\sum_{j=1}^{n} X_{ij} = 1$$

Let us assume that there are n cities for the salesman to visit. In Equation (5), the i and j indices ($i, j \in \{1, 2, \ldots, n\}$) are the numbers representing the starting and destination cities, respectively, and variable $C_{ij}$ is the cost of the salesman's traveling from city i to city j. The first constraint shows that if the salesman moves from city i to city j along a route, the $X_{ij}$ flag is equal to 1 and otherwise 0. In order to prevent the creation of sub-tours, the last two limitations of Equation (5) also state that each city can only be used once as a source and once as a destination along the optimal route [39]. To solve this problem using the CGM mechanism, each tour of the cities traveled by the salesman was considered as a position having gold, whose location coordinates were the cost of travel between pairs of cities. In other words, each of the coordinates of a location discovered to have gold, which acted as one of the variables of the optimization problem, indicated the cost of traveling between two cities. The goal was to find the location with the lowest level of impurity in the extracted gold.

### 4.1.4. Bag-of-Tasks Scheduling

Bag-of-tasks (BoT) scheduling is an important discrete problem in the field of high-performance computing that aims to reduce makespan and, consequently, increase system utilization when scheduling tasks [43]. In this problem, there are a number of tasks that should be scheduled and executed on several different physical or virtual machines. With the increases in the size and number of tasks, finding an appropriate scheduling plan is difficult and time-consuming; as a result, this problem is NP-hard. In high-performance computing and other computing systems (such as cloud computing, grid computing, and edge computing), there are some physical or virtual machines that differ in their configuration, such as processing power, memory capacity, and storage space. Furthermore, tasks have different processing, memory, and storage requirements. These differences cause a variation in the cost of task execution. The goal here is to find the best assignment of tasks to machines for the least makespan. Equation (6) shows the objective function for the BoT scheduling problem [39].

$$min \text{ Makespan} = \sum_{i=1}^{n} \sum_{j=1}^{m} C_{ij} \times X_{ij}$$

s.t.:

$$X_{ij} = \begin{cases} 1 & if \ Task_i \ is \ executed \ on \ VM_j \\ 0 & Otherwise \end{cases} \tag{6}$$

$$\sum_{j=1}^{m} X_{ij} = 1$$

Let us assume that there are n tasks that should be scheduled on m machines. In Equation (6), index i signifies the number of tasks ($i \in \{1, 2, \ldots, n\}$), and index j represents the number of machines ($j \in \{1, 2, \ldots, m\}$). The variable, $C_{ij}$, stands for the cost of executing task i on virtual machine j. The first constraint of this equation indicates that, if task i is

executed on machine j, then flag $X_{ij}$ is equal to 1 and otherwise 0. The last limitation of Equation (6) also indicates that, during the scheduling and execution of tasks, each task can only be executed on one machine [39]. To solve the BoT scheduling problem with the CGM mechanism, each sequence of scheduling was considered as a discovered position having gold, whose location coordinates were the cost of performing tasks on machines. The goal was to find the location with the minimum level of impurity in the extracted gold.

### 4.2. The Optimal Solution

There are two important parameters (number of population and number of iterations) that, if changed separately, can affect the output of all optimization methods. In order to determine the effect of each of these two parameters on the performance of methods, many experiments were performed. These experiments could determine the superiority of each method when increasing the number of population (with constant iteration) and when increasing the number of iterations (with fixed population).

#### 4.2.1. Continuous Mathematical Functions

The optimal solution value criterion was used to benchmark the mathematical functions. For evaluating continuous mathematical functions, desired values were considered for some parameters of all optimization algorithms. These parameters were the number of populations, number of variables, number of iterations, and number of experiments, and the values of 100, 50, 200, and 20 were considered for them, respectively.

We performed all experiments 20 times to ensure that the presented results had low statistical variance. Table 3 shows the best results for the optimal solution. These results showed the ability of the proposed method to find a suitable and acceptable solution for all of the mathematical functions presented in Table 2. This table indicates that the CGM algorithm was able to find the optimal solution in most cases and to discover suboptimal results for the other ones.

**Table 3.** The best results of different optimization algorithms for mathematical functions.

| Function Name | $f(x^*)$ | Number of Variables | Best Results | | | | |
|---|---|---|---|---|---|---|---|
| | | | GA | SA | PSO | IWO | CGM |
| Sphere | 0.0 | 50 | 0.208 | $8.446 \times 10^1$ | 0.067 | $8.938 \times 1001$ | 0 |
| Griewank | 0.0 | 50 | 4.851 | $2.543 \times 10^2$ | 1.195 | $3.715 \times 10^2$ | 0 |
| ZeroSum | 0.0 | 50 | 1.088 | 1.314 | 1.012 | 1.015 | 1.767 |
| Rastrigin | 0.0 | 50 | 54.27 | $5.187 \times 10^3$ | $1.784 \times 10^2$ | $5.723 \times 10^2$ | 0 |
| Qing | 0.0 | 50 | $1.601 \times 10^7$ | $2.937 \times 10^{11}$ | $1.351 \times 10^6$ | $5.372 \times 10^{11}$ | $2.847 \times 10^4$ |
| Zacharov | 0.0 | 50 | $7.781 \times 10^1$ | $8.391 \times 10^2$ | $3.514 \times 10^2$ | $2.069 \times 10^5$ | $6.511 \times 10^2$ |
| Plateau | 0.0 | 50 | $-1.52 \times 10^2$ | $-6.20 \times 10^1$ | $-2.48 \times 10^2$ | $-2.70 \times 10^1$ | $-1.16 \times 10^2$ |
| Easom | $-1.0$ | 2 | $-1.0$ | $-0.08$ | $-1.0$ | $-1.8 \times 10^{-131}$ | $-0.372$ |
| Matyas | 0.0 | 2 | 0 | 0 | 0 | 0 | 0 |

Note: $f(x^*)$ refers to the best value of the functions' solutions.

#### 4.2.2. Optimal Placement of Resources

Figure 3 depicts the results of experiments to determine the placement-of-resources problem's optimal solution using the CGM mechanism and other algorithms. For the implementation of this problem, the positions of the devices were considered randomly in the range of [−10, +10] in a ten-dimensional space. Although this ten-dimensional space is unrealistic, this hypothesis was formulated to increase the number of variables and, consequently, the problem complexity. In all experiments, the number of devices, the members of the population, and the number of iterations were different. The comparison criterion was the sum of the minimum distances between the devices and the resource in

each of the ten dimensions based on Equation (4). As seen in the figure, when compared with other algorithms, the CGM mechanism could determine the optimal solution with the least values.

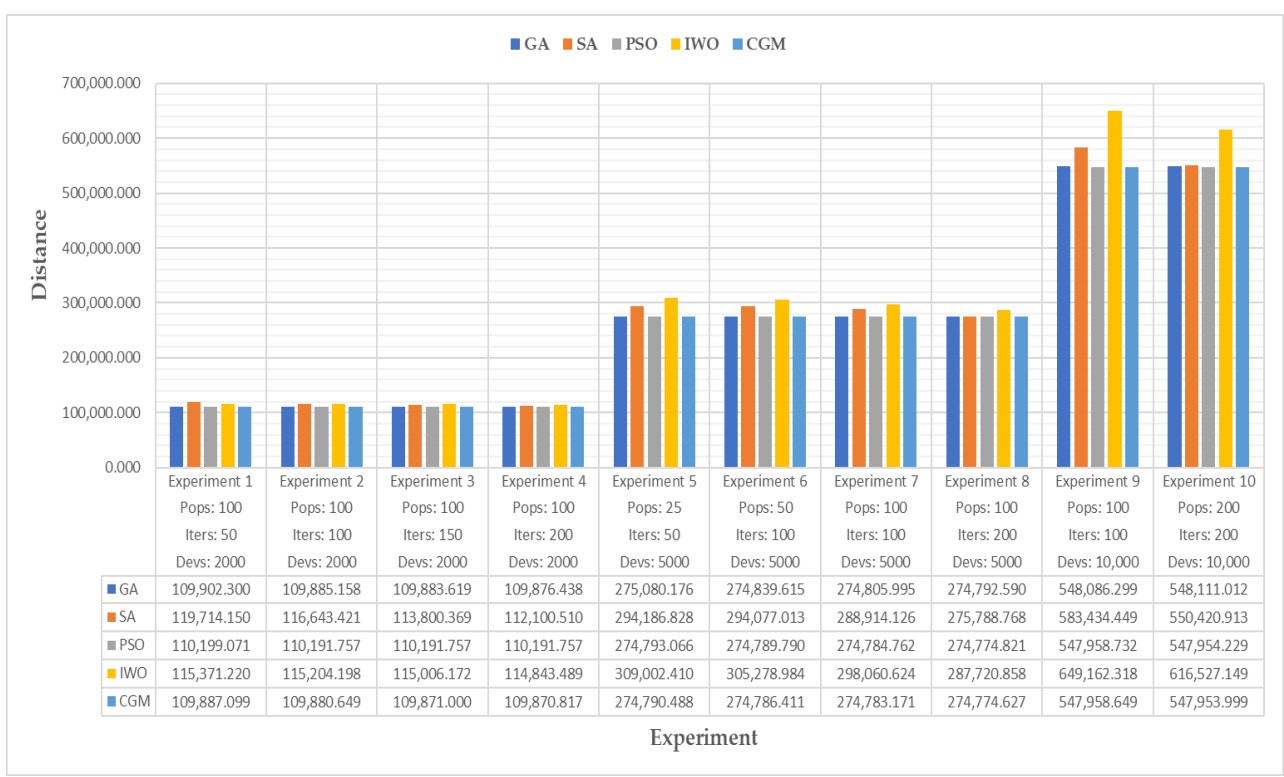

**Figure 3.** Evaluation results of the optimal-placement-of-resources problem.

In all experiments performed, the distance obtained between the resource and the devices as the optimal solution by the CGM method was shorter than the solutions determined by the other methods. The CGM method was able to improve the distance by an average of 0.004% compared with the best solutions and by an average of 8.29% compared with the worst solutions of other methods. To have a better estimation, several runs of all optimization algorithms were performed on a single configuration to determine the standard deviation, the average, and the minimum values among all algorithms. In all runs, the number of devices, the members of the population, and the number of iterations were similar and were 10,000, 50, and 100, respectively. Table 4 indicates the results of these runs for the optimal-placement-of-resources problem.

**Table 4.** Results of several runs of all optimization algorithms on a single configuration for the optimal-placement-of-resources problem in a ten-dimensional space.

| Method | Run No. 1 | Run No. 2 | Run No. 3 | Run No. 4 | Run No. 5 | Run No. 6 | Run No. 7 | Run No. 8 | Average Value | Standard Deviation | Min Value |
|---|---|---|---|---|---|---|---|---|---|---|---|
| GA | 549,710.81 | 549,791.06 | 549,774.80 | 549,761.43 | 549,643.63 | 549,685.93 | 549,597.34 | 549,784.58 | 549,718.70 | 67.13 | 549,597.34 |
| SA | 572,477.72 | 572,483.11 | 572,433.15 | 572,442.56 | 559,723.01 | 576,627.36 | 574,729.41 | 581,216.41 | 572,766.59 | 5708.25 | 559,723.01 |
| PSO | 549,512.29 | 549,535.15 | 549,535.55 | 549,523.14 | 549,527.89 | 549,512.29 | 549,529.64 | 549,526.74 | 549,525.33 | 8.46 | 549,512.29 |
| IWO | 574,240.02 | 637,875.90 | 613,579.96 | 631,532.42 | 639,198.89 | 609,705.82 | 635,726.54 | 626,909.00 | 621,096.07 | 20,456.51 | 574,240.02 |
| CGM | 549,530.68 | 549,534.87 | 549,521.60 | 549,512.29 | 549,512.28 | 549,530.93 | 549,527.60 | 549,531.06 | 549,525.16 | 8.24 | 549,512.28 |

### 4.2.3. Traveling Salesman Problem

We considered three types of datasets for implementing the traveling salesman problem on different algorithms. In the first dataset, different numbers of cities were generated with random distances between 0.1 and 10. The second dataset used the ATT48 dataset to produce distances between 48 central cities in various US states [44]. Designed to be specific to the evaluation of the algorithms developed for the traveling salesman problem,

the ATT48 dataset had a minimum tour length of 33,523. However, the third dataset employed the data presented in [45], which pertain to 15 cities. The minimum tour length of this dataset was 248.03. The criterion for comparison was the minimum cost of travel based on Equation (5). In all experiments, the members of the population and the number of iterations were selected differently. After the determination of the optimal solution to this problem using the CGM mechanism and other algorithms, Figures 4–6 present the experimental results for the three mentioned datasets, respectively.

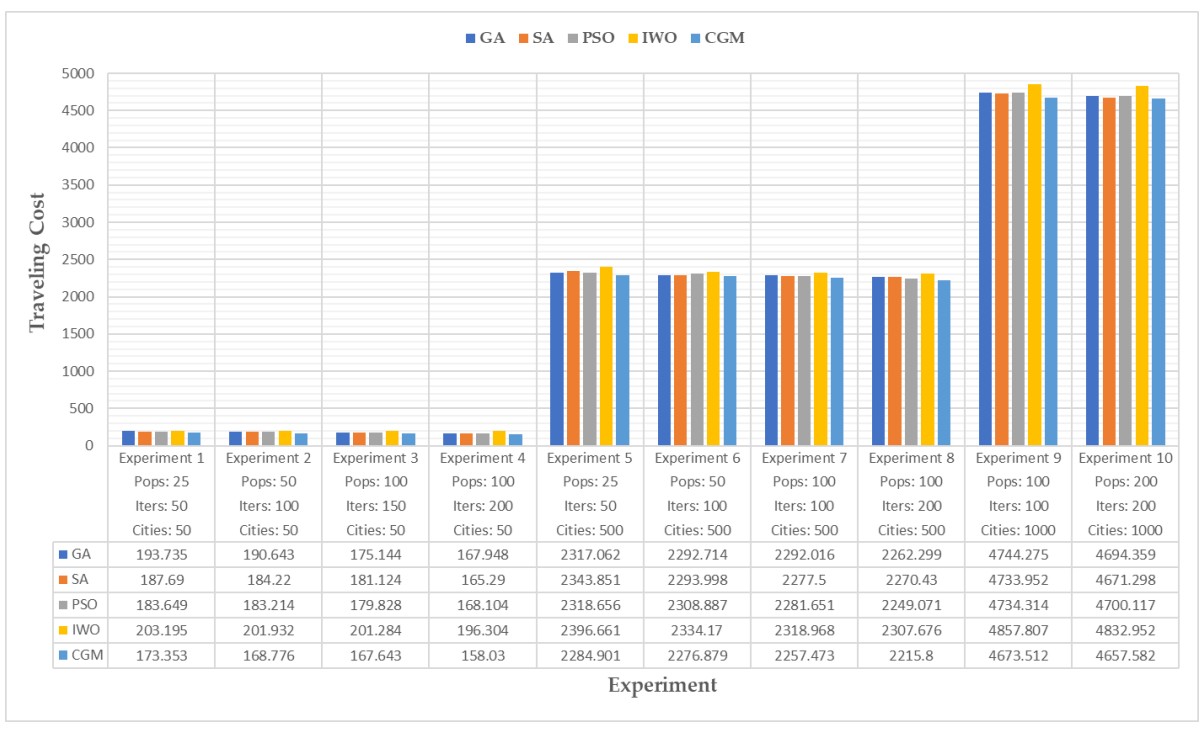

**Figure 4.** Evaluation results of the traveling salesman problem (random dataset).

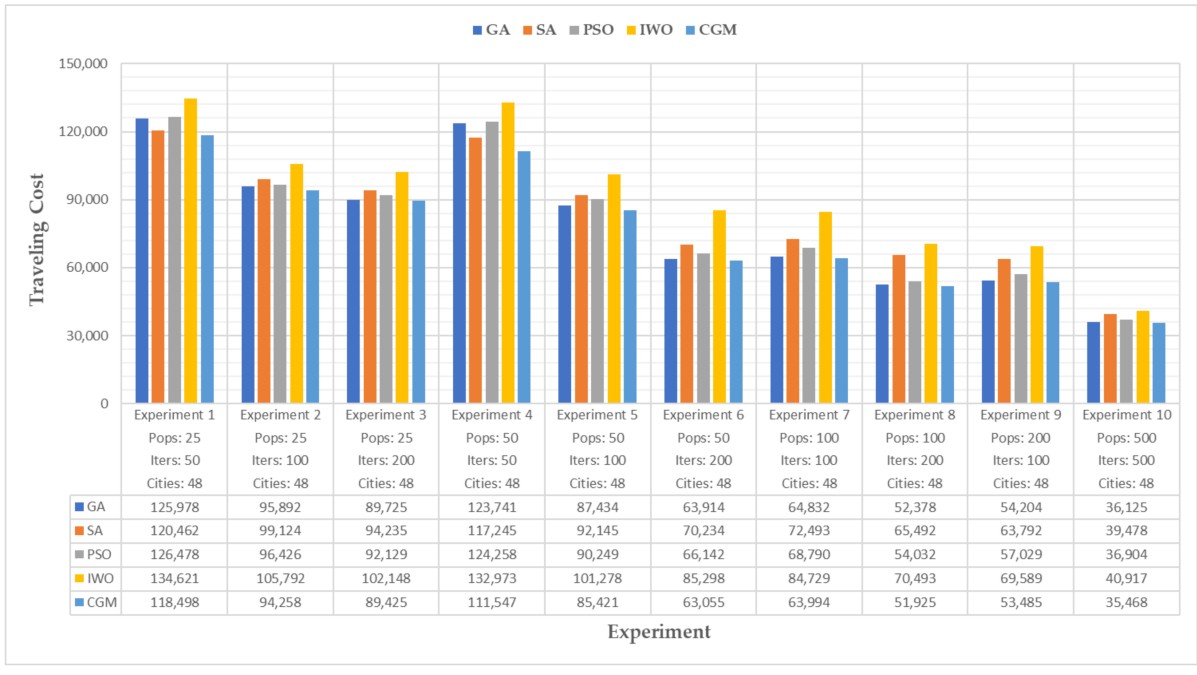

**Figure 5.** Evaluation results of the traveling salesman problem (ATT48 dataset).

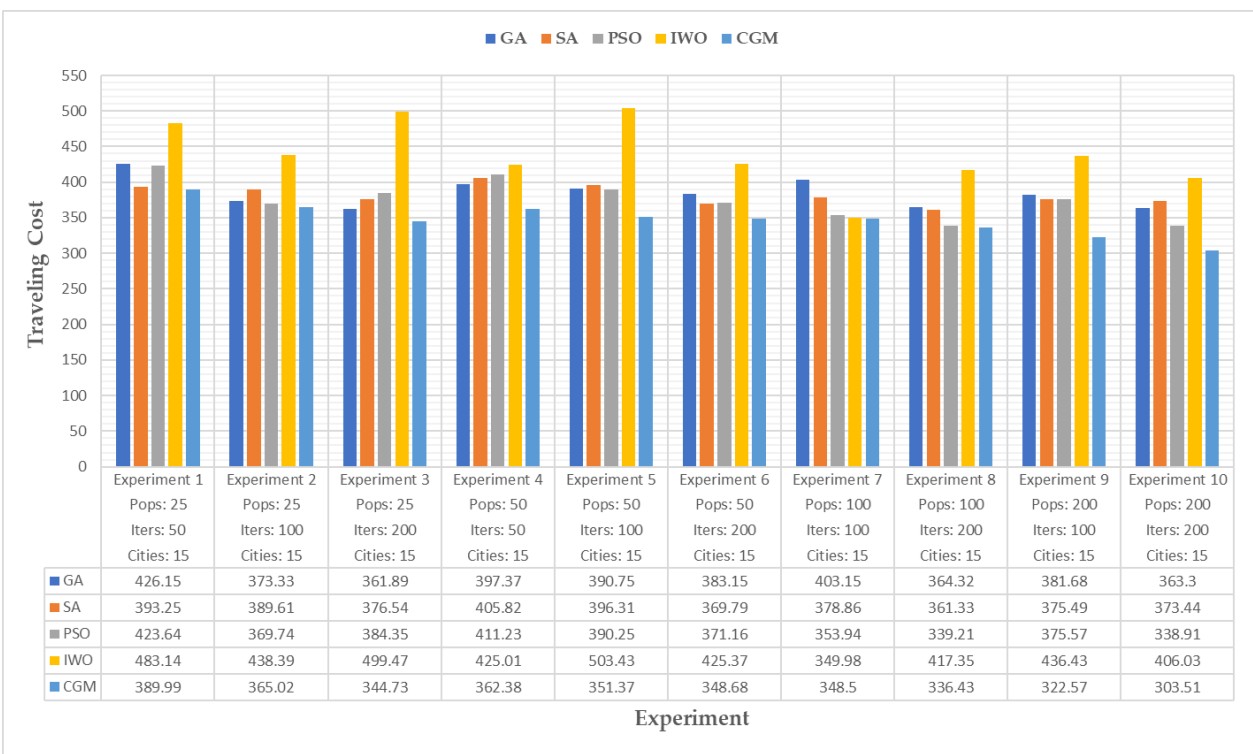

**Figure 6.** Evaluation results of the traveling salesman problem (dataset presented in [43]).

As the results show, in all experiments and for all three figures, the optimal solution in the CGM method was lower than the solutions determined by the other methods. As shown in Figure 4, the CGM method was able to improve the traveling cost by an average of 2.82% compared with the best solutions and by an average of 8.85% compared with the worst solutions of other methods. Figure 5 also shows that the CGM method was able to improve the traveling cost by an average of 1.75% and 18.05% compared with the best and worst solutions of other methods, respectively. Finally, Figure 6 indicates that the CGM mechanism was able to improve the traveling cost by an average of 5.71% compared with the best solutions and by an average of 21.42% compared with the worst solutions of other methods.

Figure 7 depicts the route between cities after the execution of the CGM mechanism in the developed simulator for the ATT48 dataset (iterations 1 and 300). In this figure, the vertices show the location of the cities, and the edges represent the route between the two selected cities. To have a better estimation, several runs of all optimization algorithms were performed on a single configuration for the traveling salesman problem to determine the standard deviation, the average, and the minimum values among all algorithms. In all runs, the number of cities was 1000 with random distances between 0.1 and 10, and the members of the population and the number of iterations were similar and were 150 and 250, respectively. Table 5 indicates the results of these runs.

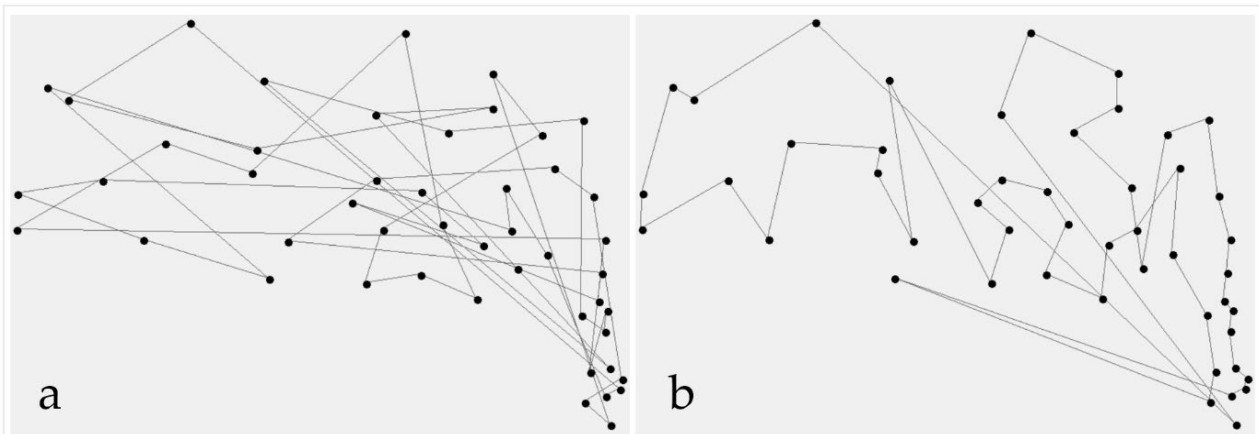

**Figure 7.** Route between cities after the execution of the CGM mechanism for ATT48 dataset: (**a**) iteration No. 1 and (**b**) iteration No. 300.

**Table 5.** Results of several runs of all optimization algorithms on a single configuration for the traveling salesman problem.

| Method | Run No. 1 | Run No. 2 | Run No. 3 | Run No. 4 | Run No. 5 | Run No. 6 | Run No. 7 | Run No. 8 | Average Value | Standard Deviation | Min Value |
|---|---|---|---|---|---|---|---|---|---|---|---|
| GA | 4712.09 | 4687.92 | 4741.16 | 4727.21 | 4721.43 | 4656.99 | 4738.43 | 4696.38 | 4710.20 | 26.68 | 4656.99 |
| SA | 4712.01 | 4744.90 | 4713.08 | 4733.56 | 4673.23 | 4720.70 | 4721.07 | 4688.33 | 4713.36 | 21.66 | 4673.23 |
| PSO | 4750.67 | 4710.09 | 4720.07 | 4734.73 | 4703.22 | 4751.52 | 4749.81 | 4705.28 | 4728.17 | 19.69 | 4703.22 |
| IWO | 4858.97 | 4840.58 | 4848.44 | 4902.71 | 4800.24 | 4887.21 | 4869.01 | 4850.01 | 4857.14 | 29.08 | 4800.24 |
| CGM | 4704.46 | 4681.68 | 4713.34 | 4687.17 | 4690.89 | 4656.66 | 4701.17 | 4656.29 | 4686.46 | 19.70 | 4656.29 |

### 4.2.4. Bag-of-Tasks Scheduling

To implement the BoT scheduling problem, the number of tasks, the number of virtual machines, the number of population members, and the number of iterations were different; furthermore, the number of instructions for each task and the processing power of each virtual machine were randomly selected. The comparison criterion was the minimum makespan based on Equation (6). Figure 8 presents the results of experiments for determining the optimal solution of this problem using the CGM mechanism and other algorithms. As the results show, in all experiments performed, the makespan determined by the CGM method was shorter than the makespan obtained by the other methods. The CGM method was able to improve the makespan by an average of 1.17% compared with the best solutions and by an average of 12.43% compared with the worst solutions of other methods.

Figure 9 also depicts how tasks were assigned to virtual machines after running the CGM method in the developed simulator. In this figure, the horizontal axis represents the task number, and the vertical axis shows the virtual machine number. Each of the points indicates the assignment of a task to a VM.

To have a better estimation, several runs of all optimization algorithms were performed on a single configuration of the BoT scheduling problem to determine the standard deviation, the average, and the minimum values among all algorithms. In all runs, the number of tasks, the number of virtual machines, the members of the population, and the number of iterations were similar and were 10,000, 200, 80, and 100, respectively. Table 6 indicates the results of these runs for the BoT scheduling problem.

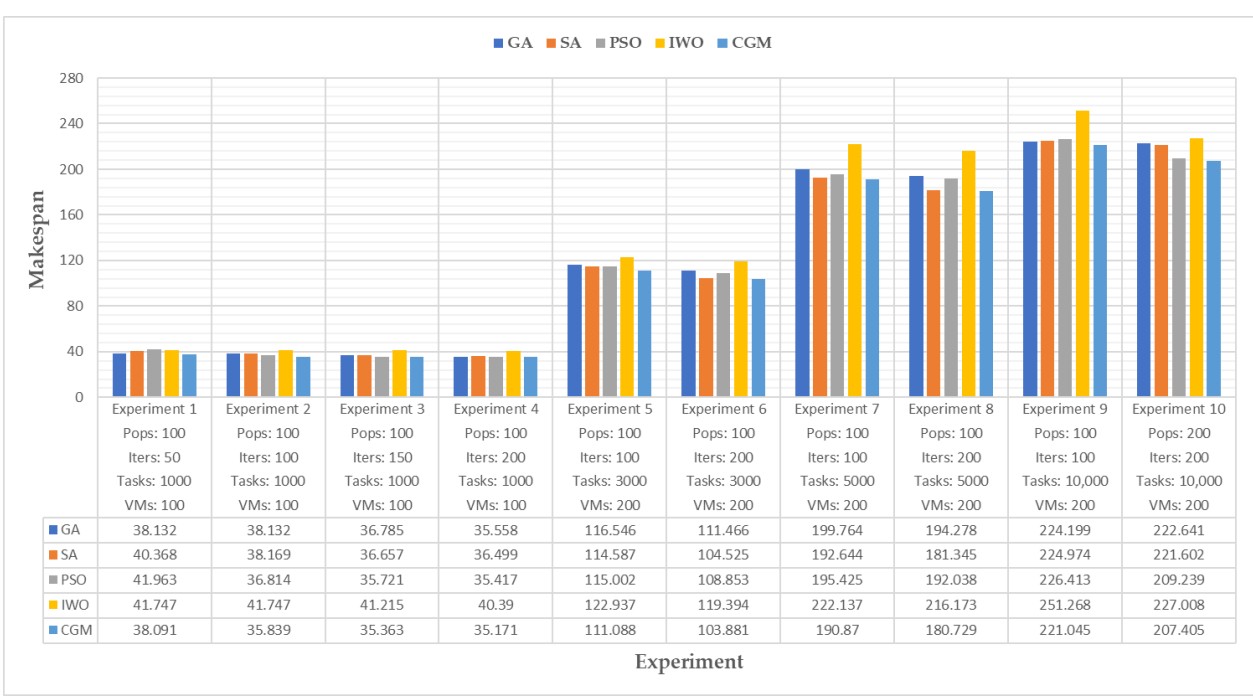

**Figure 8.** Evaluation results of the BoT scheduling problem.

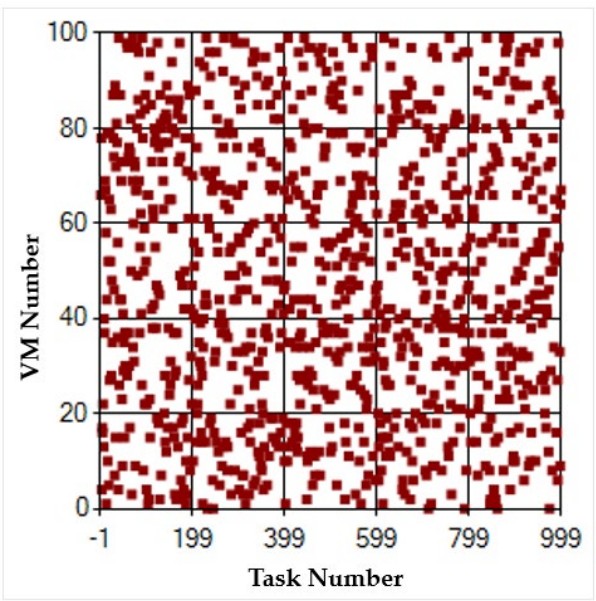

**Figure 9.** Assignment of tasks to virtual machines after executing the CGM method on the developed simulator (for experiment 4).

**Table 6.** Results of several runs of all optimization algorithms on a single configuration for the BoT scheduling problem.

| Method | Run No. 1 | Run No. 2 | Run No. 3 | Run No. 4 | Run No. 5 | Run No. 6 | Run No. 7 | Run No. 8 | Average Value | Standard Deviation | Min Value |
|--------|-----------|-----------|-----------|-----------|-----------|-----------|-----------|-----------|---------------|--------------------|-----------|
| GA | 233.195 | 226.14 | 227.44 | 238.01 | 230.35 | 231.97 | 236.77 | 237.70 | 232.62 | 4.28 | 226.14 |
| SA | 229.523 | 233.63 | 229.76 | 236.34 | 228.44 | 239.50 | 241.29 | 227.18 | 233.73 | 5.02 | 227.18 |
| PSO | 230.911 | 240.17 | 231.55 | 238.14 | 233.56 | 228.27 | 232.60 | 240.02 | 234.90 | 4.22 | 228.27 |
| IWO | 259.89 | 279.42 | 267.81 | 278.21 | 258.65 | 276.45 | 271.80 | 280.47 | 273.26 | 8.28 | 258.65 |
| CGM | 229.153 | 233.16 | 226.46 | 235.74 | 225.58 | 221.25 | 228.28 | 236.48 | 229.56 | 4.94 | 221.25 |

All the results in Sections 4.2.1–4.2.4 indicate the capability of the CGM mechanism in determining the best optimal solutions for some mathematical functions and practical problems (including continuous and discrete) compared with other algorithms. Hence, the CGM mechanism can be applied to applications in different fields and provides better optimization results. In the following, we review other performance metrics for the proposed method.

### 4.3. Convergence

One of the criteria for evaluating optimization algorithms is convergence. When the number of population members, the number of iterations, and the size of the problem are considered constant, the algorithm that can converge to the best possible solution in the least number of iterations is known as a fast convergence algorithm. For both continuous and discrete problems, and even with a small population, the CGM mechanism has quasi-optimal solutions from the very first iterations and quickly converges to the optimal solution. In other words, the optimal solution can be reached with a small population and the least possible number of iterations. As a result, this method can be used in devices with limited computational, storage, and energy resources that need to determine the result in the shortest possible time. Figures 10–12 show the convergence speed of the CGM in comparison with other optimization algorithms in different problems. In these figures, the number of members of the population was 20, and the number of iterations was 50.

The results presented in Figure 10 for the optimal-placement-of-resources problem show that the CGM method was able to converge to the minimum distance by the 13th iteration. Among other benchmark algorithms, the PSO algorithm could converge to the same result by the 27th iteration, and the others could not achieve this value until the end of 50 iterations. The results presented in Figure 11 for the traveling salesman problem indicate that the CGM mechanism could converge to the minimum traveling cost by the 24th iteration. However, other benchmark algorithms could not converge to this value until the end of 50 iterations. In addition, the results presented in Figure 12 for the BoT scheduling problem also demonstrate that the CGM method could converge to the minimum makespan by the 39th iteration. However, other benchmark algorithms could not achieve this solution until the end of all iterations. The same result was obtained in cases with a larger number of members.

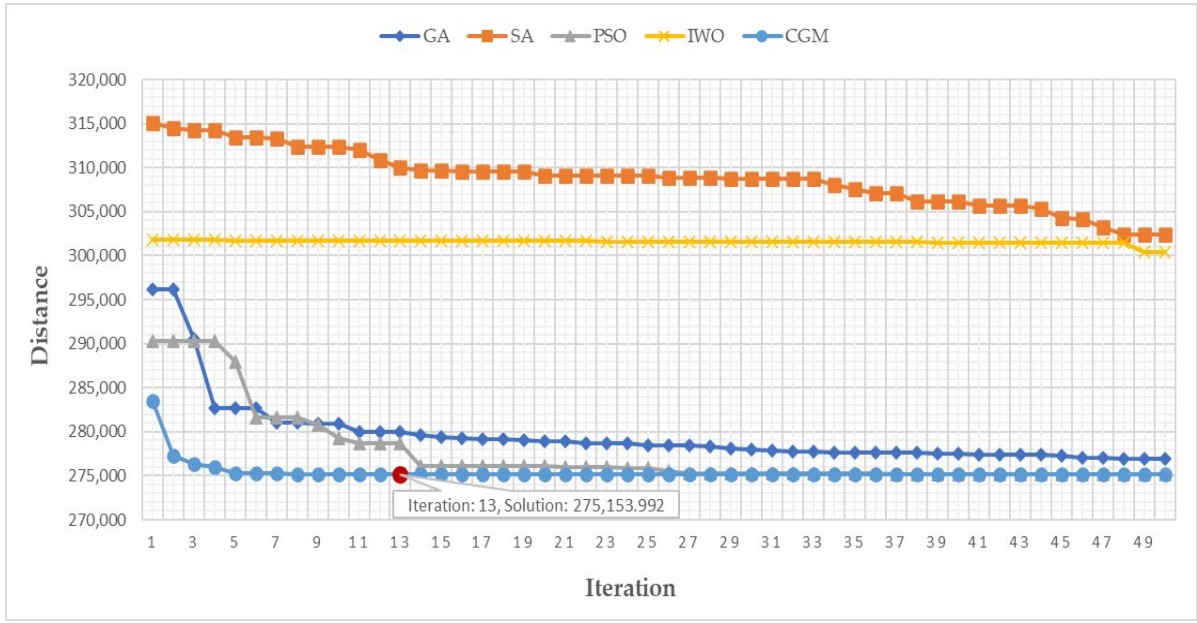

**Figure 10.** Investigation of convergence in different algorithms for the optimal-placement-of-resources problem (the number of devices was 5000 in a 10-dimensional space).

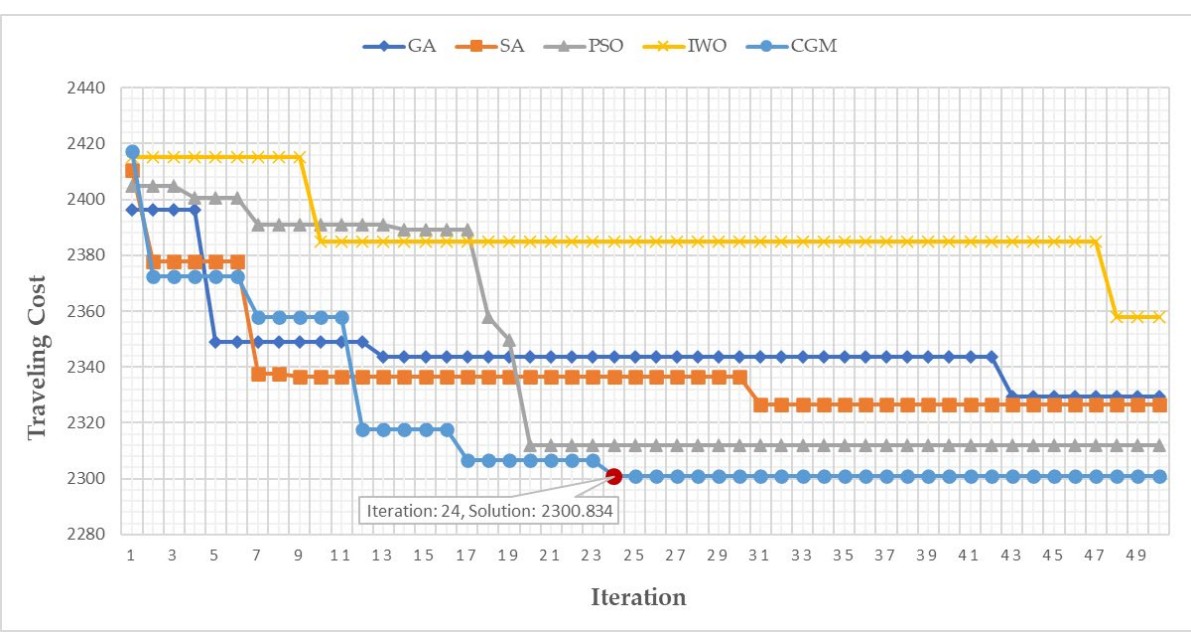

**Figure 11.** Investigation of convergence in different algorithms for the traveling salesman problem (the number of cities was 500).

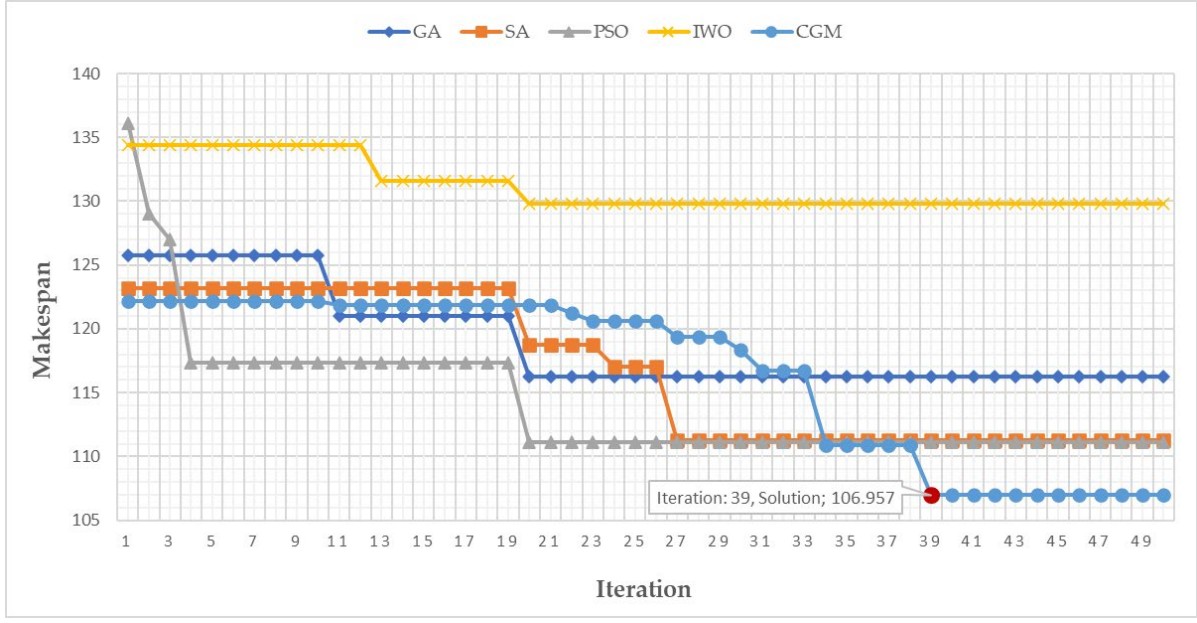

**Figure 12.** Investigation of convergence in different algorithms for the BoT scheduling problem (the number of tasks was 5000, and the number of computational resources was 200).

*4.4. Scalability*

Assuming that the number of population members and the number of iterations of the algorithm are kept constant, if the size of the problem increases, the CGM mechanism has acceptable scalability compared with other algorithms. In order to evaluate the scalability of the proposed method in terms of the optimal solution value, several experiments were performed with a wide range of different problem sizes. The results of these experiments can be seen in Figures 13–15. In all three figures, the number of members of the population was 20, and the number of iterations was 100.

Figure 13 shows that on average, in all experiments, and for different problem sizes, the CGM method was able to improve the distance by 0.03% compared with the best

solutions and by 13.44% compared with the worst solutions of other methods. Figure 14 also shows that on average, in all experiments, and for different problem sizes, the CGM method was able to improve the travel cost by 1.08% compared with the best solutions and by 4.94% compared with the worst solutions of other methods. Figure 15 also shows that on average, in all experiments, and for different problem sizes, the CGM method was able to improve the distance by 2.61% compared with the best solutions and by 16% compared with the worst solutions of other methods. As a result, it can be said that the CGM method has better scalability.

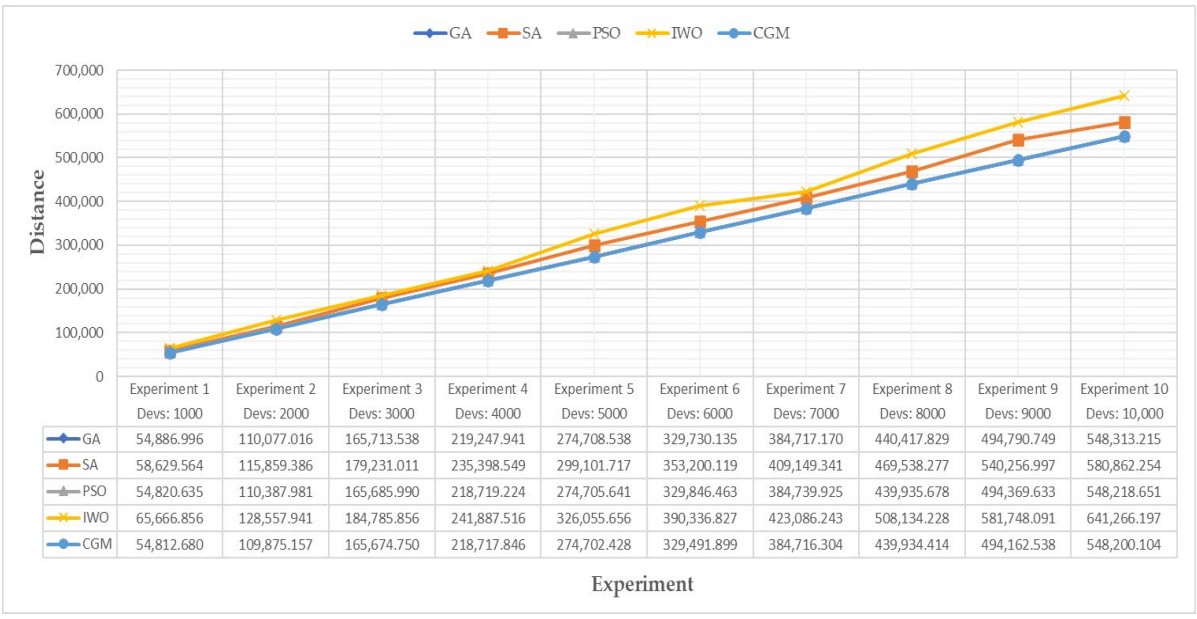

**Figure 13.** Scalability of the CGM mechanism for the optimal-placement-of-resources problem compared with other algorithms.

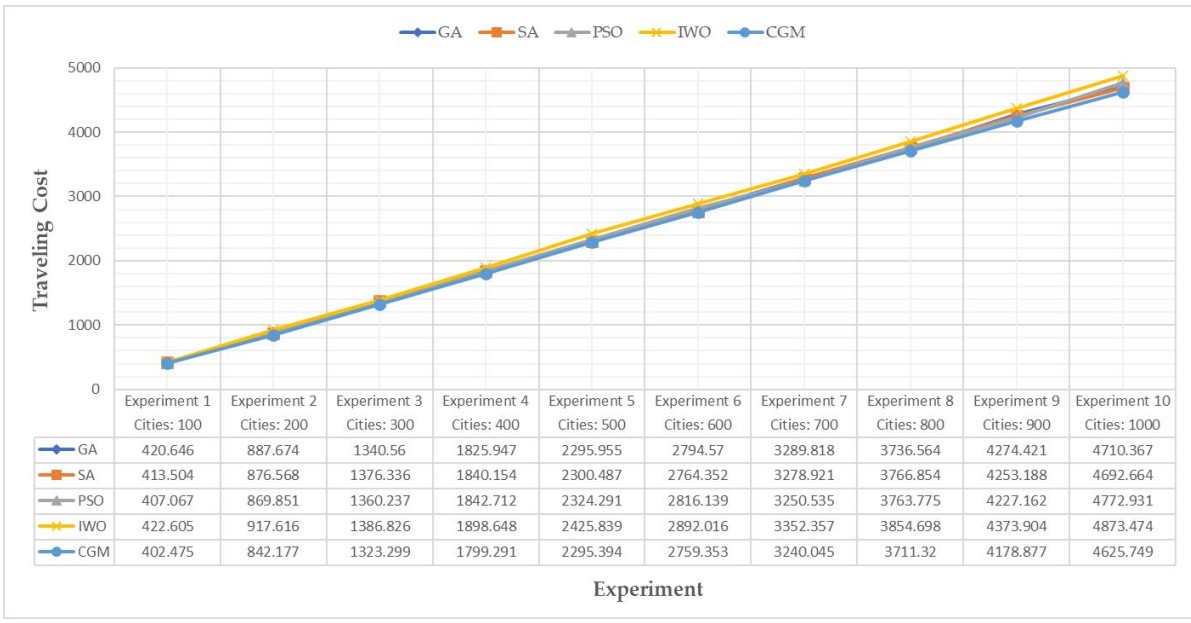

**Figure 14.** Scalability of the CGM mechanism for the traveling salesman problem compared with other algorithms.

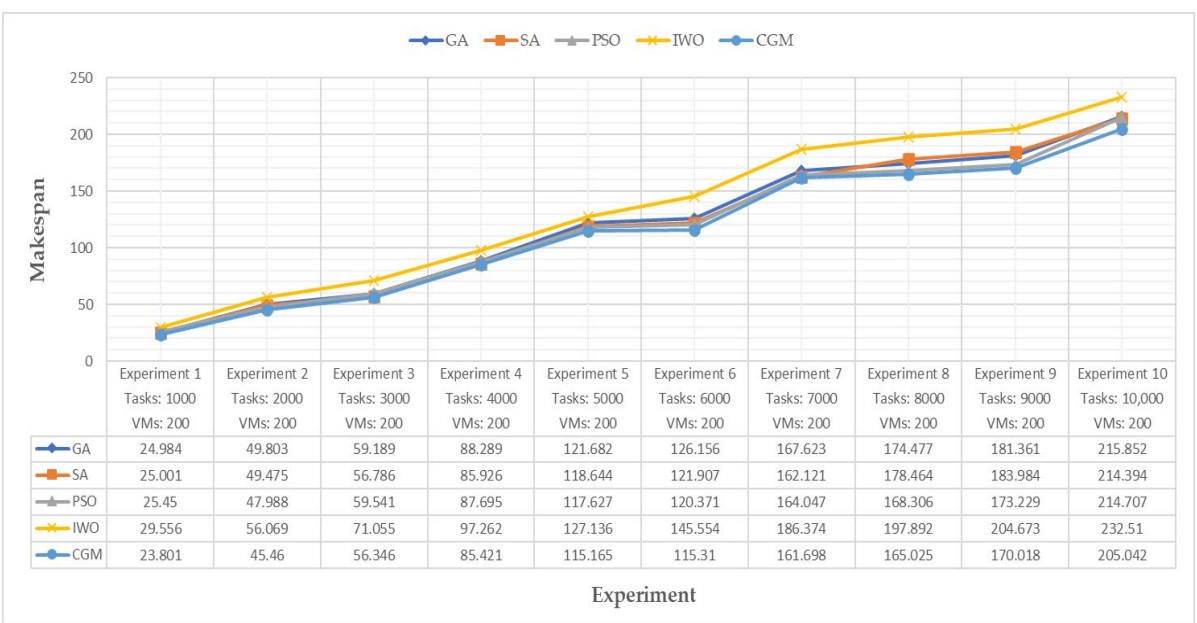

**Figure 15.** Scalability of the CGM mechanism for the BoT scheduling problem compared with other algorithms.

### 4.5. Search Space

In the CGM mechanism, although the best values are always determined during the iterations of the algorithm, and based on these best values, the three positions of first-best, second-best, and third-best are updated, miners always try to explore and extract the entire search space and avoid the optimal solution (first-best position). As a result, unlike most existing optimization methods in which agents move toward the optimal solution during iterations, the CGM mechanism always examines the entire search space to find better solutions. Figures 16–18 show the results of the search space for different optimization problems. As shown in these three figures, the number of members of the population was 5, and the number of iterations was 50. This advantage is due to the use of random variable $\varphi$. Therefore, according to the contents of Section 3.3, in continuous problems, the exploration process is performed in the range of the values of the best positions; therefore, the next points with gold are close to the best points (as shown in Figure 16, after iteration 21, with which the best solution is found, miners try to explore around this point to find more gold). On the other hand, because of the possibility of selecting $\varphi$ in the entire search space, the process of exploration in discrete problems covers the entire search space. This can be seen in Figures 17 and 18, where after finding the best solution, miners try to find more gold in the entire search space.

### 4.6. Computational Demand

Another criterion that can affect the performance of an optimization algorithm is execution time. Generally, the execution time is calculated from the moment the algorithm starts until the end of all iterations. The CGM mechanism is able to find the optimal solution for mathematical functions and continuous practical problems in the shortest possible time. Figures 19–21 illustrate this metric for the optimal-placement-of-resources problem, the traveling salesman problem, and the BoT scheduling problem, respectively. As can be seen in these figures, the computational demand of the CGM method for continuous practical problems is acceptable compared with other methods.

For discrete practical problems, the execution time required by the CGM mechanism is almost the same as the execution time required by other methods. This computational demand depends on the type of discrete problem. Only in discrete problems where duplicate values are not attributable to variables (such as the traveling salesman problem,

where it is not possible to re-select a city), when the number of variables, the number of population members, and the number of iterations of the algorithm increase simultaneously, the required execution time also increases. As a result, the efficiency of the algorithm is reduced from the computational perspective, and it might not be possible to use this algorithm in the real-time applications of discrete problems.

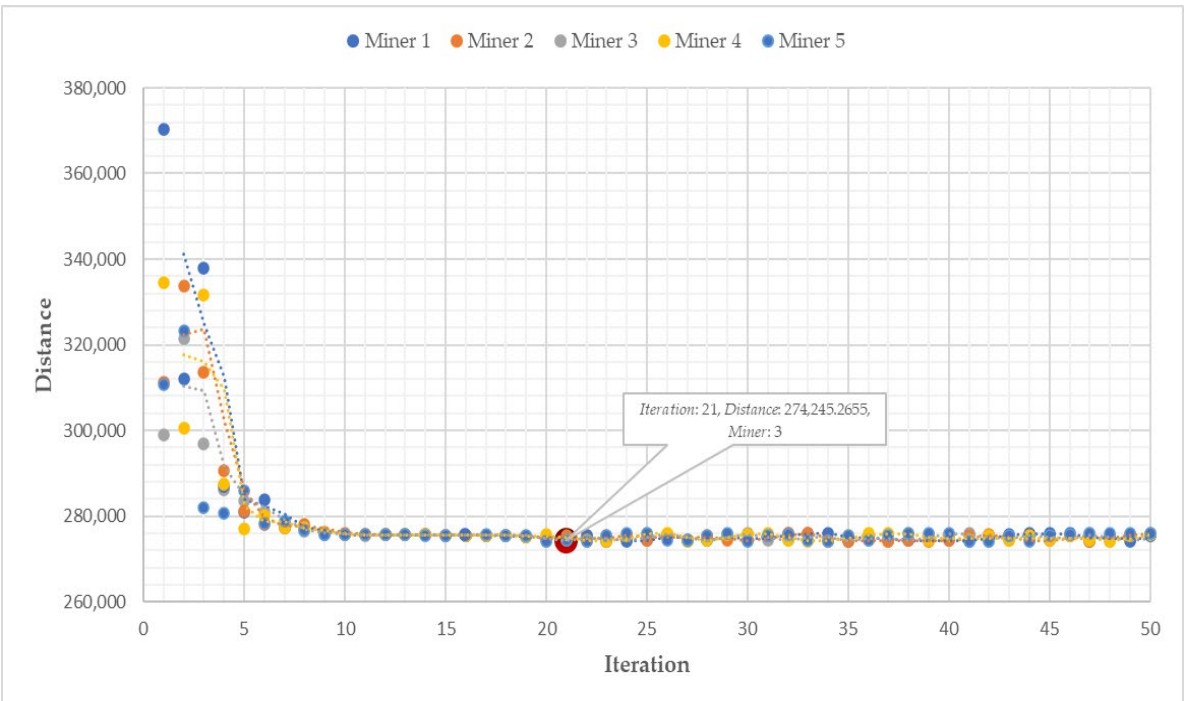

**Figure 16.** Collaboration of miners to determine the appropriate solution for the optimal-placement-of-resources problem around locations with the possibility of finding the optimal solution (the number of devices was 5000 in a 10-dimensional space).

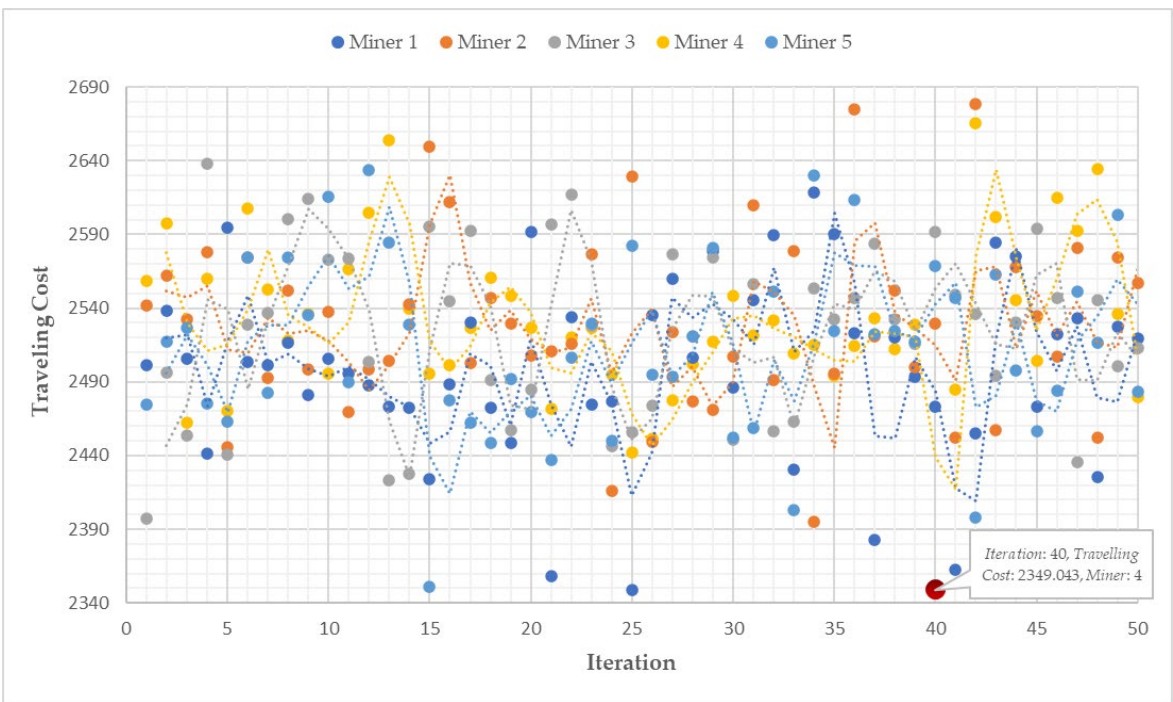

**Figure 17.** Collaboration of miners to determine the appropriate solution for the traveling salesman problem in the entire search space (the number of cities was 500).

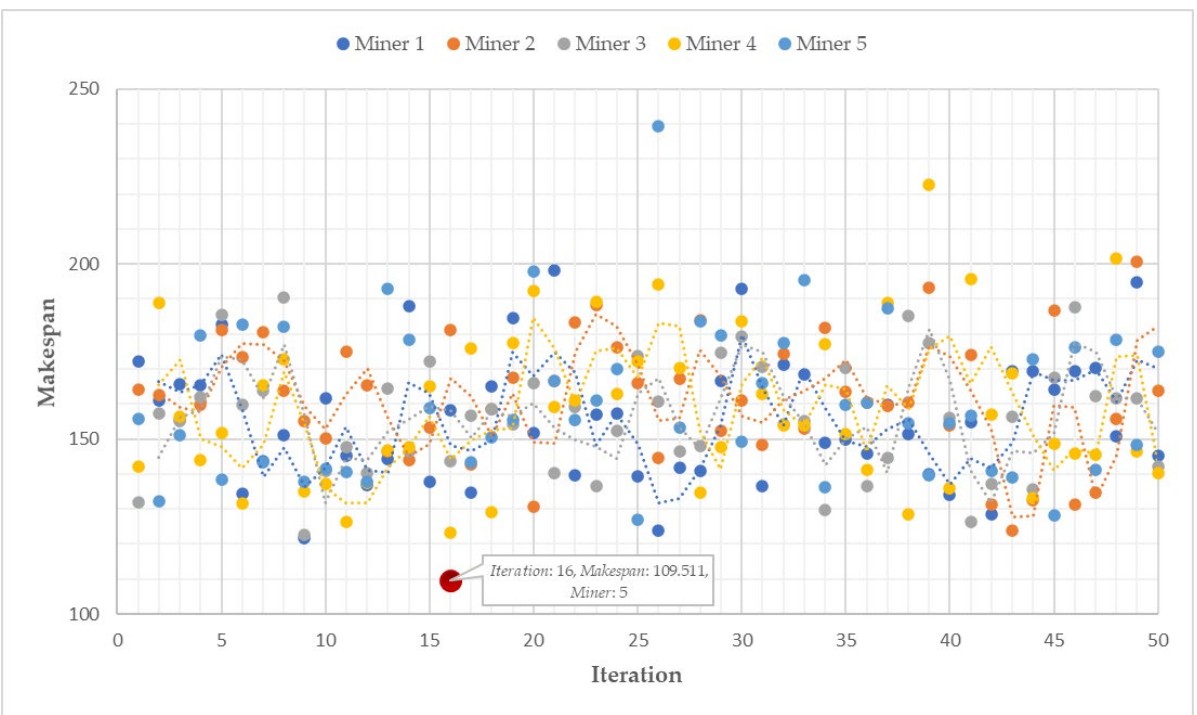

**Figure 18.** Collaboration of miners to determine the appropriate solution for the BoT scheduling problem in the entire search space (the number of tasks was 5000, and the number of computational resources was 200).

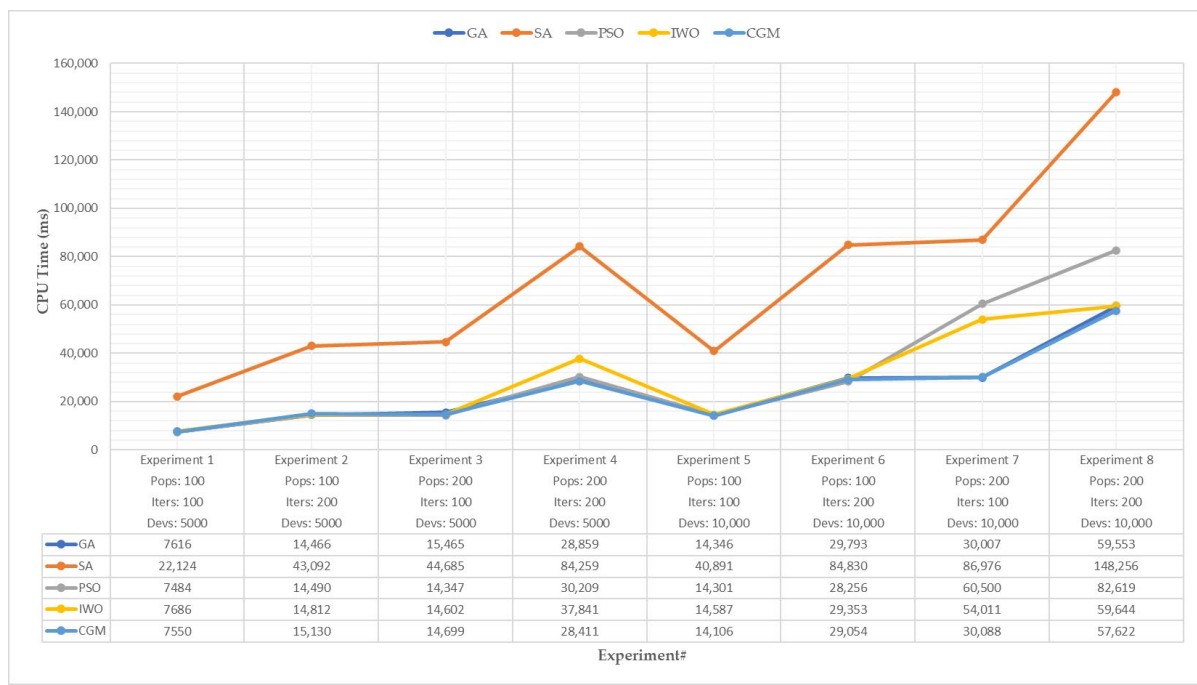

**Figure 19.** Evaluation results of execution time (CPU time) criterion for the optimal-placement-of-resources problem with large parameters (in milliseconds).

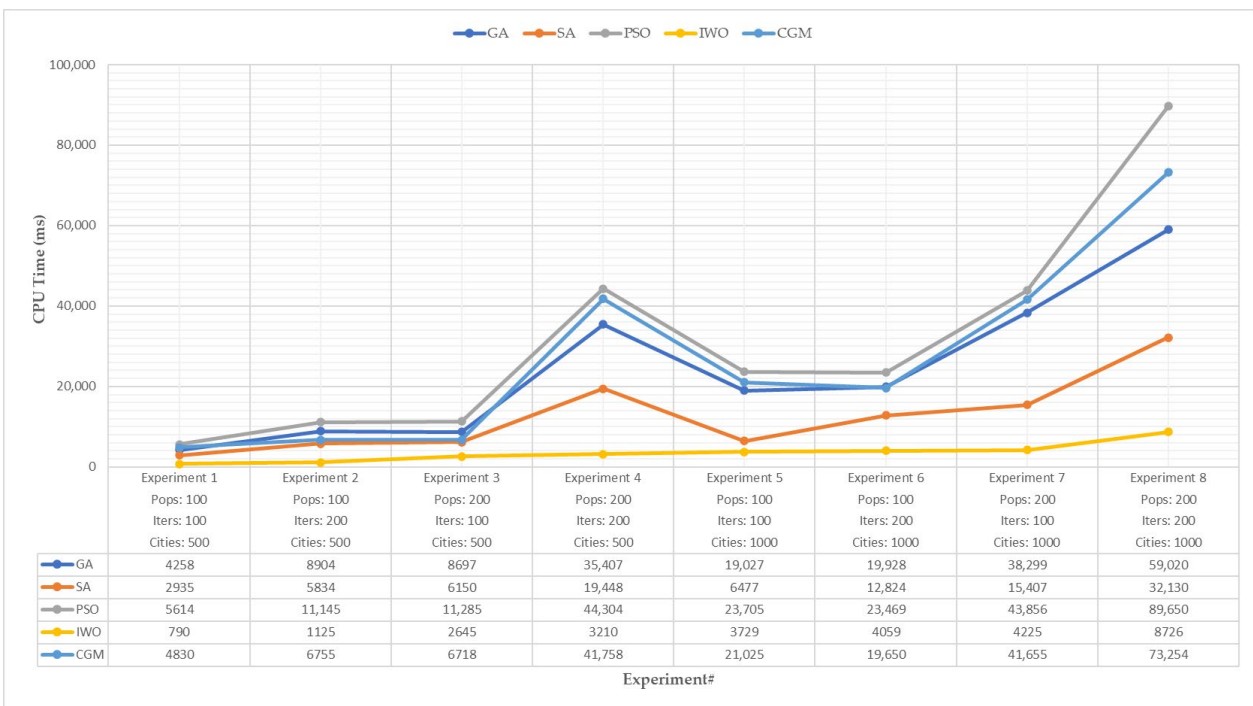

**Figure 20.** Evaluation results of execution time (CPU time) criterion for the traveling salesman problem with large parameters (in milliseconds).

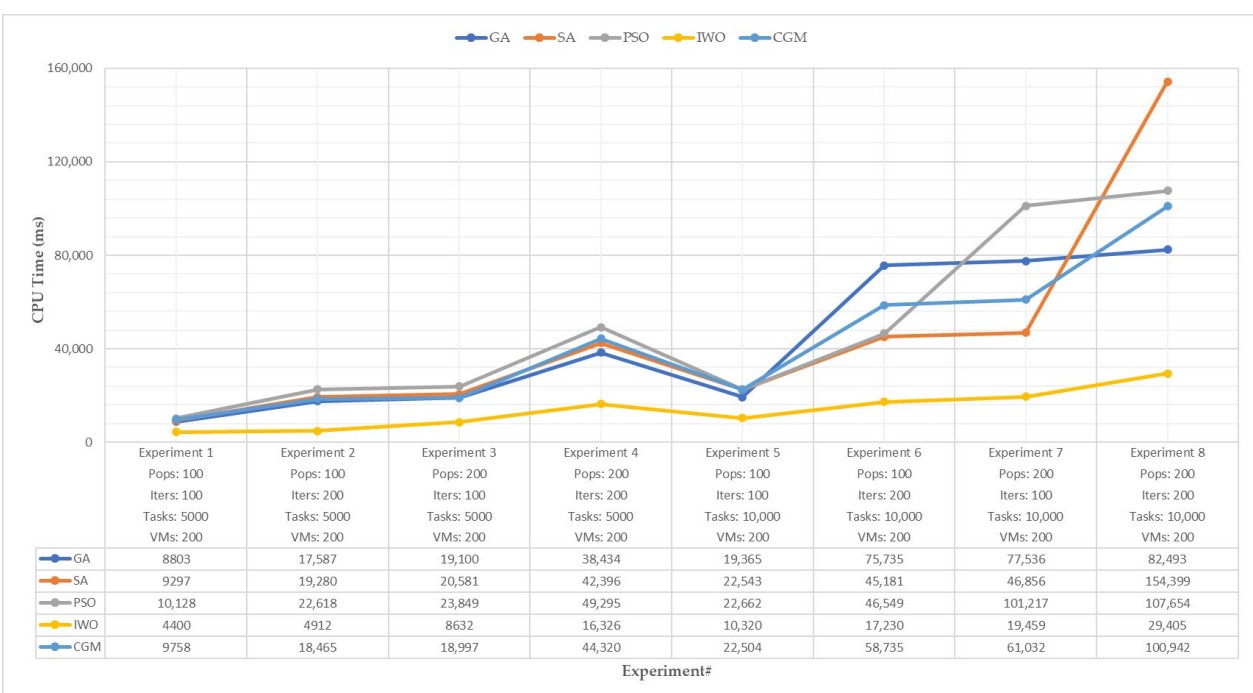

**Figure 21.** Evaluation results of execution time (CPU time) criterion for the BoT scheduling problem with large parameters (in milliseconds).

## 5. Discussion

In practical terms, there are no optimization algorithms that can solve all the problems with the best possible solution [46]. In terms of well-known criteria, such as optimal solution, convergence, scalability, search space, and computational demand, the CGM mech-

anism has promising performance for both continuous and discrete evaluated problems, and its results are comparable to those of other well-known methods.

The CGM algorithm was developed in such a way that depending on the type of problem (continuous or discrete), it has a well-defined behavior and does not need converting continuous problems to discrete problems (and vice versa) or converting data. The transparent operation of this algorithm is such that it can be easily used to develop and solve a continuous or discrete optimization problem. Moreover, the experimental results indicate that the proposed method has a comparable, and in some cases better, performance in solving all continuous and discrete problems. Unlike most existing optimization mechanisms, in which algorithm settings must be obtained by trial-and-error and changes are based on problem type, the CGM mechanism does not require a continuous change of parameters, and as shown in all evaluations, the optimization problem can be solved with the same default values considered for the base algorithm. The CGM mechanism can solve minimization and maximization problems without many changes.

In this algorithm, since the positions of the miners are considered as initial solutions to the problem, their positions should be randomly dispersed to cover the geographical area and not limit the search operation to a specific part of the geographic space. In the beginning, if the miners are located near each other in one part of the geographical area, the CGM algorithm is only able to find the local optimal solution where the miners are and not the global optimal solution, which is the main target of the problem. Therefore, the location of the initial population should be placed throughout the whole space of the problem as a fine-tuning precondition for the correct execution of the CGM mechanism.

## 6. Conclusions

Inspired by the process of gold exploration and exploitation, this paper simulates the behavior of gold miners and introduces a new natural optimization method called collaborative gold mining (CGM). Compared with some famous optimization methods, such as GA, SA, PSO, and IWO, the results of the evaluations show that the proposed mechanism, while finding the optimal solution, has a good performance in terms of some other criteria, such as convergence, scalability, search space, and computational demand. The fewer the number of iterations required to determine the quasi-optimal or optimal solution to an optimization algorithm is, the shorter the response time is. Consequently, the CGM algorithm can be employed to solve real problems in different application domains, especially on devices with limited resources. To evaluate the efficiency of the proposed CGM mechanism, several mathematical functions and three application examples, including the optimal placement of resources, traveling salesman problem, and bag-of-tasks scheduling, were selected and examined.

Future work will focus more on utilizing the CGM algorithm in other applications, especially real-time applications, and evaluating its performance against other mechanisms. Another avenue to explore is modifying the CGM algorithm so that it can solve combinatorial optimization problems. The initialization of the three best variables has a great impact on the output of the proposed algorithm. So, another research direction is to examine how to choose the initial position for the three best variables exactly, in such a way that the maximum amount of profit is obtained for the gold miner.

**Author Contributions:** A.S. and B.J. have contributed to the study conception and design, and text preparation, dataset collection, code implementation, and analysis were performed by all authors. All authors have read and agreed to the published version of the manuscript.

**Funding:** This research received no external funding.

**Institutional Review Board Statement:** Not applicable.

**Informed Consent Statement:** Not applicable.

**Data Availability Statement:** Not applicable.

**Conflicts of Interest:** The authors declare no conflict of interest.

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
