# Peer review of "Collaborative Gold Mining Algorithm: An Optimization Algorithm Based on the Natural Gold Mining Process"

_electronics, doi:10.3390/electronics11223824_

Round 1
Reviewer 1 Report
Inspired by the process of gold exploration and exploitation, a new metaheuristic and stochastic optimization algorithm called collaborative gold mining (CGM) is proposed in this paper. This paper is well written and numerical comparison is provided in detail. This reviewer has the following comments:
1) The motivation of your proposal should be highlighted based on the critical of published algorithms. In the current version, only one sentence placed in lines 159-161 is not enough.
2) In the comparison with GA, SA, PSO, IWO, CGM, the state-of-the-art version of these algorithms should be cited rather than the self-coded traditional version.
3) There is no cross within the global optimal solution of TSP, thus the solution b in Figure 7 (iterations 300) is far away from the global optimal due to the cross of edges. Based on this observation and the number of cities is only 48, the performance of CGM cannot be fully established.
4) The performance improvement compared with the best benchmark is not significant for the majority of experiments.
Reviewer 2 Report
The authors have presented a new optimisation technique which seems to be promising. However the work has to be revised to make it more sound and authentic. The following are the comments:
1. None of the optimisation techniques can claim to work well for all the optimisation problems. So the results have to be taken with the cost. The comparison of the proposed optimisation technique with other well-established techniques such as genetic algorithm, PSO, etc., cannot be generalised. So the authors are suggested to carry out a several runs of optimisation on the same problem, find the standard deviation as well the average and the minimum values. This will give a better estimate than 1 single experiment on each problem.
2. Authors are also suggested to carry out the estimation of computational complexity. Some of the well know algorithms are very quick is finding the optimal solution and the time taken to run them is also less helping their real world utility.
3. There are several other test functions that are commonly used to test the function. The authors can refer to some recent publications on optimisation techniques
4. How does the initial starting point effect the overall output? Was there any fine tuning incorporated to select the initial point?
Reviewer 3 Report
In my opinion, the introduction should be further improved to better specify the motivation of the work and the characteristics of the methodology. As it is, it is not clear how it works. Additionally, the author should state clearly in which aspect this work extends state of the art, i.e., highlight the novelty…
Also, highlight the research gap in existing research and in literature. The importance of the proposed integrated approach with respect to the problem statement should have been in focus.
At the end of related works, highlight better in some lines what overall technical gaps are observed in existing works that led to the design of the proposed approach. To better delineate the context and the different possible solutions, you can consider the following papers as references: https://www.sciencedirect.com/science/article/pii/S0957417421012598 and https://link.springer.com/chapter/10.1007/978-3-030-99079-4_19.
The future scope of the methodology should be extended/highlighted. Improve the conclusion, and clarify the conclusion of this article with its significance for follow-up research
Round 2
Reviewer 1 Report
All the revisions should be highlighted in the revised manuscript.
Author Response
Thanks for all your valuable comments, the revised manuscript with all highlighted revisions was attached here.

Reviewer 2 Report
Dear Authors,
Thanks for the revision which clarifies the novelty of the work. However a few more improvements are needed:
thanks for the revision which clarifies the novelty of the work. However a few more improvements are needed:
1. Computationally the method appears to be more intensive than other methods. The authors are requested to add a few lines to explain this and also the use cases where the proposed method may not be applicable: for example real-time applications.
